# Phase 1 double-blind randomized safety trial of the Janus kinase inhibitor tofacitinib in systemic lupus erythematosus

Sarfaraz A. Hasni[1✉], Sarthak Gupta[1,2], Michael Davis[1], Elaine Poncio[1], Yenealem Temesgen-Oyelakin[1], Philip M. Carlucci[2], Xinghao Wang[2], Mohammad Naqi[1], Martin P. Playford[3], Rishi R. Goel[2], Xiaobai Li[4], Ann J. Biehl[5], Isabel Ochoa-Navas[1], Zerai Manna[1], Yinghui Shi[6], Donald Thomas[7], Jinguo Chen[8], Angélique Biancotto[8], Richard Apps[8], Foo Cheung[8], Yuri Kotliarov[8], Ashley L. Babyak[8], Huizhi Zhou[8], Rongye Shi[8], Katie Stagliano[8], Wanxia Li Tsai[6], Laura Vian[6], Nathalia Gazaniga[6], Valentina Giudice[9], Shajia Lu[6], Stephen R. Brooks[10], Meggan MacKay[11], Peter Gregersen[11], Nehal N. Mehta[3], Alan T. Remaley[12], Betty Diamond[11], John J. O'Shea[13], Massimo Gadina[6] & Mariana J. Kaplan[2]

Increased risk of premature cardiovascular disease (CVD) is well recognized in systemic lupus erythematosus (SLE). Aberrant type I-Interferon (IFN)-neutrophil interactions contribute to this enhanced CVD risk. In lupus animal models, the Janus kinase (JAK) inhibitor tofacitinib improves clinical features, immune dysregulation and vascular dysfunction. We conducted a randomized, double-blind, placebo-controlled clinical trial of tofacitinib in SLE subjects (ClinicalTrials.gov NCT02535689). In this study, 30 subjects are randomized to tofacitinib (5 mg twice daily) or placebo in 2:1 block. The primary outcome of this study is safety and tolerability of tofacitinib. The secondary outcomes include clinical response and mechanistic studies. The tofacitinib is found to be safe in SLE meeting study's primary endpoint. We also show that tofacitinib improves cardiometabolic and immunologic parameters associated with the premature atherosclerosis in SLE. Tofacitinib improves high-density lipoprotein cholesterol levels ($p = 0.0006$, CI 95%: 4.12, 13.32) and particle number ($p = 0.0008$, CI 95%: 1.58, 5.33); lecithin: cholesterol acyltransferase concentration ($p = 0.024$, CI 95%: 1.1, −26.5), cholesterol efflux capacity ($p = 0.08$, CI 95%: −0.01, 0.24), improvements in arterial stiffness and endothelium-dependent vasorelaxation and decrease in type I IFN gene signature, low-density granulocytes and circulating NETs. Some of these improvements are more robust in subjects with *STAT4* risk allele.

A full list of author affiliations appears at the end of the paper.

The pathogenesis of systemic lupus erythematosus (SLE) involves dysregulation of multiple innate and adaptive immune pathways[1]. The role of the innate immune system, specifically type I interferons, low-density neutrophils, and neutrophil extracellular traps (NETs), is now recognized as a potential fundamental player in SLE pathogenesis and its associated vascular damage[2]. The risk of cardiovascular disease (CVD) and premature atherosclerosis is significantly increased in SLE, particularly in young women[3,4]. Efforts to modulate SLE vasculopathy, atherosclerosis development, and progression have been unsuccessful[5].

The immune dysregulation characteristic of SLE has been proposed to play prominent roles in driving premature atherosclerosis[6]. Specifically, the type I-Interferon (IFN) pathway and a distinct subset of neutrophils called low-density granulocytes (LDGs) that display enhanced ability to form neutrophil extracellular traps (NETs) have been directly implicated in vascular damage, atherosclerotic plaque formation, and CVD progression[7,8].

The Janus kinase/signal transducers and activators of transcription (JAK/STAT) pathway is a fundamental signaling cascade used by cells to respond to a wide variety of cytokines and growth factors[9]. Many inflammatory cytokines implicated in SLE pathogenesis, including type I and II interferons (IFNs), signal through a JAK-STAT pathway[10,11]. JAK inhibitors (jakinibs) have displayed efficacy in various murine models of lupus[12]. Clinical trials using the jakinibs showed clinical efficacy in arthritis in patients with mild-to-moderate SLE[13,14]. Furthermore, tofacitinib modulated dysregulated neutrophil and type I IFN responses, endothelial dysfunction, and lipoprotein profiles in murine lupus[12].

We hypothesized that mitigating the aberrant activation of innate immune pathways with a JAK inhibitor could lead to the improvement in cardiometabolic parameters and immune dysregulation associated with premature vascular damage in SLE. To this end, we conducted a phase Ib/IIa, double-blind, placebo-controlled clinical trial of tofacitinib in SLE subjects with mild-to-moderate disease. As the presence of the *STAT4* risk allele (rs7574865) has been associated with more severe clinical phenotype and significantly increased risk of vascular disease in SLE, and since type I IFNs activate *STAT4*, we stratified subjects based on the presence (+) or absence (−) of *STAT4* risk allele which has been suggested to increase the production of and sensitivity to type I IFNs in peripheral blood mononuclear cells of SLE patients, to investigate the effect(s) of these haplotypes on the clinical and immunologic response to tofacitinib[15–17].

Here, we show that tofacitinib is safe and well-tolerated in subjects with mild-to-moderate SLE. There are no unexpected adverse events or worsening of SLE disease activity, with no severe adverse events, opportunistic infections, or thromboembolic events with the use of tofacitinib. The tofacitinib treatment improves high-density lipoprotein cholesterol levels and particle numbers, lecithin: cholesterol acyltransferase concentration, and cholesterol efflux capacity. It also improves arterial stiffness and endothelium-dependent vasorelaxation. The tofacitinib use significantly decreased the type I IFN gene signature, levels of low-density granulocytes, and circulating NETs. Some of these changes were more robust in SLE subjects with *STAT4* risk allele. Long-term studies are needed to determine the efficacy of tofacitinib in CVD prevention in SLE.

## Results

**Safety and disease activity**. Thirty SLE subjects were enrolled, all completed the study and were included in analyses. Baseline subject demographic and clinical characteristics stratified by their

*STAT4* risk allele status are shown in Table 1 and Supplementary Fig. 1. There were significantly more African American subjects in the *STAT4* risk allele-negative subgroup in both tofacitinib and placebo groups ($P = 0.002$). The subjects with *STAT4* risk allele were younger and mostly Hispanic, but these differences were not statistically significant. There were 71 adverse events: 43 in the tofacitinib group and 28 in the placebo group, with no serious AEs in the tofacitinib group; the differences in AE were not statistically significant (Table 2). Most of the AEs observed in the tofacitinib group were mild (16/43) and moderate (5/43) upper respiratory infections that either self-resolved or after treatment with oral antibiotics. No herpes zoster reactivation, BK viremia, or venous thromboembolic events were recorded. There were no clinical or statistically significant changes observed in the tofacitinib group compared to baseline measurements and to the placebo group in other laboratory safety parameters (Table 3). As compared to placebo in the tofacitinib group, the hemoglobin difference in change score was −0.33 (95% CI −0.33, −0.88) at day 56, −0.20 (95% CI, −0.81, 0.40) at day 84; white blood cell count difference in change score was −0.63 (95% CI −0.63, −1.46) at day 56, −0.52 (95% CI, −1.35, 0.30) at day 84; absolute neutrophil count difference in change score was −0.58 (95% CI −0.58, −1.35) at day 56, −0.21 (95% CI, −0.85, 0.43) at day 84; platelet count difference in change score was −15.36 (95% CI −15.36, −35.79) at day 56, 5.3 (95% CI, −12.33, 22.93) at day 84; serum AST difference in change score was 0.71 (95% CI, 0.71, −5.14) at day 56, −13.8 (95% CI, −34.91, 7.31) at day 84; and serum ALT difference in change score was 1.58 (95% CI, 1.58, −4.55) at day 56, −2.15 (95% CI, −8.86, 4.56) at day 84 (Table 3). None of these differences were statistically significant. None of the patients in either group met the disease flare criteria during the trial, and there were no new BILAG 2004 A or B scores. The baseline mean SLEDAI 2 K score in the tofacitinib group was 5.1 ± 2.2 mean ± standard deviation, compared to 5.5 ± 3.7 in the placebo group; the difference in change scores (tofacitinib vs. placebo) was 0.04 (95% CI: −1.04, 1.11) at day 56 and −0.72 (95% CI: −1.98, 0.53) at day 84. The baseline mean BILAG 2004 score in the tofacitinib group was 7.6 ± 4.6 mean ± standard deviation, compared to 9.3 ± 4.3 in the placebo group; the difference in change scores (tofacitinib vs. placebo) was 1.56 (95% CI: −1.86, 4.98) at day 56 and −2.04 (95% CI: −4.96, 0.89) at day 84. The baseline mean PGA score in the tofacitinib group was 0.8 ± 0.8 (mean ± standard deviation), compared to 1.2 ± 0.9 in the placebo group; the difference in change in scores (tofacitinib vs. placebo) was 0.18 (95% CI: −0.28, 0.64) at day 56 and 0.23 (95% CI: −0.30, 0.76) at day 84. None of these differences were statistically significant. The SLE serological disease activity (complement C3 and C4 levels) and the patient-reported outcomes (SF-36) were similar at baseline visit and did not have a significant difference in change scores at day 56 or day 84 (Table 4). There was a numerical increase in anti-ds-DNA antibody titers in both groups during the study, but these increases were not statistically significant. This study was not powered to assess clinical efficacy.

Despite randomization, there were some differences between the tofacitinib and the placebo group at the baseline visit. The subjects on placebo reported a statistically significant higher degree of fatigue, as measured by MD-Fatigue scale at baseline, and a significant improvement in fatigue during the study. The subjects on placebo had higher baseline DAS-28-ESR and lower baseline anti-ds-DNA levels as compared to the tofacitinib group, but these were not statistically significant differences (Table 4). The safety and disease activity data of subjects on tofacitinib and placebo stratified based on presence or absence of *STAT4* risk allele revealed non-statistically significant differences in some variables between groups (except for as noted elsewhere); the

**Table 1 Baseline demographic, clinical, and vascular characteristics of SLE subjects enrolled in the trial based on their *STAT4* risk allele status.**

| | Tofacitinib | | Placebo | | Total *N* = 30) | *P* value at baseline |
| --- | --- | --- | --- | --- | --- | --- |
| | *STAT4* risk allele present (*n* = 10) | *STAT4* risk allele absent (*n* = 10) | *STAT4* risk allele present (*n* = 5) | *STAT4* risk allele absent (*n* = 5) | | |
| Age (years) mean ± SD | 44 ± 16.9 | 53.8 ± 13.9 | 35.6 ± 15.9 | 44.4 ± 7.2 | 45.9 ± 15.2 | 0.16 |
| *Race/ethnicity*, N (%) | | | | | | |
| African American | 1 (10%) | 8 (80%) | 0 (0%) | 4 (80%) | 13 (43.3%) | 0.002** |
| Asian | 0 (0%) | 1 (10%) | 0 (0%) | 1 (20%) | 2 (6.7%) | |
| Caucasian | 2 (20%) | 0 (0%) | 0 (0%) | 0 (0%) | 2 (6.7%) | |
| Hispanic | 7 (70%) | 1 (10%) | 5 (100%) | 0 (0%) | 13 (43.3%) | |
| *Gender*, N (%) | | | | | | 0.70 |
| Female | 8 (80%) | 9 (90%) | 5 (100%) | 4 (80%) | 26 (86.7%) | |
| Male | 2 (20%) | 1 (10%) | 0 (0%) | 1 (20%) | 4 (13.3%) | |
| Disease duration, mean ± SD | 10.8 ± 11.61 | 15.6 ± 11.42 | 9.8 ± 12.44 | 16.2 ± 9.68 | 13.1 ± 11.1 | 0.65 |
| SLEDAI 2 K, mean ± SD | 5.2 ± 2.7 | 4.9 ± 1.79 | 5.6 ± 4.34 | 5.4 ± 3.44 | 5.2 ± 2.7 | 0.97 |
| Prednisone, *N* (%) | 6 (60%) | 3 (30%) | 2 (40%) | 3 (60%) | 14 (46.7%) | 0.52 |
| Prednisone dose, mean ± SD | 5.0 ± 2.7 | 5.0 ± 4.3 | 7.5 ± 3.5 | 6.67 ± 2.9 | 5.7 ± 3.0 | 0.73 |
| HCQ, *N* (%) | 9 (90%) | 10 (100%) | 5 (100%) | 5 (100%) | 29 (96.67%) | 0.56 |
| Rt CAVI, mean ± SD | 7.37 ± 1.08 | 8.24 ± 1.49 | 6.9 ± 1.62 | 7.1 ± 1.02 | 7.54 ± 1.36 | 0.22 |
| PWV, m/s | 7.46 ± 1.61 | 7.4 ± 2.82 | 7.22 ± 2.15 | 6.98 ± 0.7 | 7.32 ± 2.00 | 0.98 |
| Ln RHI | 0.43 ± 0.35 | 0.68 ± 0.25 | 0.83 ± 0.13 | 0.88 ± 0.28 | 0.65 ± 0.32 | 0.026* |
| Anti-dsDNA antibody, *N* (%) | 1 (10%) | 3 (30%) | 2 (40%) | 1 (20%) | 7 (23.33%) | 0.56 |
| ACA IgG, *N* (%) | 4 (40%) | 1 (10%) | 0 (0%) | 1 (20%) | 6 (20%) | 0.22 |
| ACA IgM, *N* (%) | 6 (60%) | 1 (10%) | 2 (40%) | 0 (0%) | 9 (30%) | 0.04* |
| Lupus anticoagulant, *N* (%) | 5 (50%) | 4 (40%) | 2 (40%) | 1 (20%) | 12 (40%) | 0.74 |
| History of lupus nephritis, *N* (%) | 2 (20%) | 3 (30%) | 1 (20%) | 0 (0%) | 6 (20%) | 0.60 |

*SLEDAI 2 K* Systemic Lupus Erythematosus Disease Activity Index 2000, *HCQ* hydroxychloroquine, *Rt. CAVI* right cardioankle vascular index, *PWV*, pulse wave velocity, *Ln RHI* Log Reactive Hyperemia Index, *Anti-ds-DNA* anti-double-stranded DNA antibody, *ACA IgG* anticardiolipin antibody IgG, *ACA IgM* anticardiolipin antibody IgM.
*p < 0.05, **p<0.01.
The baseline variables were compared using the mean values using analysis of variance (ANOVA) for continuous variables and Chi-square test for categorical variables.
Two-tailed tests were used where appropriate.
No adjustments were made for multiple comparisons.

clinical relevance of these differences is uncertain and needs to be further explored in a larger sample (Supplementary Tables 1 and 2).

**Tofacitinib modulated the type I IFN gene signature and pSTAT levels**. To assess target engagement in subjects receiving tofacitinib, we measured phosphorylation of STAT1, STAT3, and STAT5. Tofacitinib use led to significant inhibition of pSTAT1 in CD4$^+$ T cells at days 14 and 56 of the trial ($P = 0.023$; Fig. 1a). There was also a trend observed towards nonsignificant inhibition of pSTAT1 in subjects on a placebo that is of unknown significance and without any known plausible biological explanation. In contrast, pSTAT3, pSTAT5, and pSTAT1 levels were not significantly different in CD8$^+$ T cells, CD20$^+$ B cells or CD14$^+$ monocytes between tofacitinib or placebo-treated patients (Fig. 1b and Supplementary Fig. 2a, b). Assessment of these parameters one month after drug discontinuation showed that the effects on pSTAT1 did not persist after drug was stopped.

We next assessed gene expression between groups at baseline, days 56 and 84 by whole blood RNAseq. In total, 90 genes were found to be twofold different (22 up and 68 down) by ANOVA comparison between tofacitinib and placebo treatment on day 56. Of the 68 genes that were significantly downregulated in the tofacitinib group at day 56, 19 were Interferon-Stimulated Genes (ISG). By day 84, the levels of most of these 19 ISGs had returned to pretreatment levels, but a few genes were still significantly downregulated (Fig. 1c and Supplementary Fig. 3). Nanostring

was used to verify the impact of tofacitinib on the IFN signature (Fig. 1c). The IFN response gene score was not significantly different at baseline between subjects on tofacitinib and placebo. At day 56, the subjects on tofacitinib had a significant reduction in levels of ISGs in comparison to placebo ($P = 0.01$), which remained significant at the day 84 visit ($P = 0.02$) (Supplementary Fig. 4)[18].

**Tofacitinib modulated dysregulated neutrophil responses**. The JAK/STAT pathway and cognate cytokines are crucial in neutrophil biology[19]. We examined the impact of tofacitinib on LDGs (a distinct subset of proinflammatory neutrophils elevated in SLE) and their ability to synthesize NETs. We found that there was a significant reduction in the percentage of LDGs in SLE subjects on tofacitinib compared to placebo at day 56 ($P = 0.048$) and the decrease was sustained at day 84 ($P = 0.014$; Fig. 2a), whereas total neutrophils did not significantly decrease during active treatment (Fig. 2b and Table 4). There was also a concomitant increase in LDGs in the placebo group which partially explains this statistically significant difference. While the role of NETs in promoting inflammation and vascular damage in SLE has been previously described in vitro and in vivo, the effect of the *STAT4* risk allele in NET formation is unknown[7]. We found that SLE subjects positive for the *STAT4* risk allele had significantly higher levels of circulating NET complexes at baseline when compared to *STAT4* risk allele-negative subjects ($P = 0.02$) (Fig. 2c). The levels of circulating NET complexes significantly

**Table 2 Adverse event frequency and severity.**

| Body system preferred term severity | Tofacitinib adverse events = 43 | Placebo adverse events = 28 | Total adverse events = 71 | P value |
|---|---|---|---|---|
| *Any event* | | | | 0.09 |
| Mild | 32 (74.4%) | 23 (82.1%) | 55 (77.5%) | |
| Moderate | 11 (25.6%) | 3 (10.7%) | 14 (19.7%) | |
| Severe | 0 (0%) | 2 (7.1%) | 2 (2.8%) | |
| *Cardiac disorders* | | | | NA |
| Mild | 1 (2.3%) | 0 (0.0%) | 1 (1.4%) | |
| Moderate | 0 (0.0%) | 0 (0.0%) | 0 (0.0%) | |
| Severe | 0 (0.0%) | 0 (0.0%) | 0 (0.0%) | |
| *Eye disorders* | | | | 1.00 |
| Mild | 1 (2.3%) | 3 (10.7%) | 4 (5.6%) | |
| Moderate | 0 (0.0%) | 1 (3.6%) | 1 (1.4%) | |
| Severe | 0 (0.0%) | 0 (0%) | 0 (0.0%) | |
| *Gastrointestinal disorders* | | | | 1.00 |
| Mild | 4 (9.3%) | 4 (14.3%) | 8 (11.3%) | |
| Moderate | 2 (4.6%) | 1 (3.6%) | 3 (4.2%) | |
| Severe | 0 (0.0%) | 0 (0%) | 0 (0.0%) | |
| *General disorders* | | | | NA |
| Mild | 2 (4.6%) | 1 (3.6%) | 3 (4.2%) | |
| Moderate | 0 (0.0%) | 0 (0.0%) | 0 (0.0%) | |
| Severe | 0 (0.0%) | 0 (0.0%) | 0 (0.0%) | |
| *Infections* | | | | 0.55 |
| Mild | 16 (37.2%) | 4 (14.3%) | 20 (28.2%) | |
| Moderate | 5 (11.6%) | 0 (0.0%) | 5 (7.0%) | |
| Severe | 0 (0%) | 0 (0.0%) | 0 (0.0%) | |
| *Investigations (clinical laboratory abnormalities)* | | | | 1.00 |
| Mild | 2 (4.7%) | 2 (7.1%) | 4 (5.6%) | |
| Moderate | 1 (2.3%) | 0 (0%) | 1 (1.4%) | |
| Severe | 0 (0%) | 1 (3.6%) | 1 (1.4%) | |
| *Metabolism disorders* | | | | NA |
| Mild | 0 (0%) | 0 (0.0%) | 0 (0.0%) | |
| Moderate | 1 (2.3%) | 0 (0.0%) | 1 (1.4%) | |
| Severe | 0 (0%) | 0 (0.0%) | 0 (0.0%) | |
| *Musculoskeletal disorders* | | | | NA |
| Mild | 1 (2.3%) | 2 (7.1%) | 3 (4.2%) | |
| Moderate | 0 (0%) | 0 (0.0%) | 0 (0.0%) | |
| Severe | 0 (0%) | 0 (0.0%) | 0 (0.0%) | |
| *Nervous system disorders* | | | | 0.42 |
| Mild | 4 (9.3%) | 7 (25.0%) | 11 (15.5%) | |
| Moderate | 1 (2.3%) | 0 (0.0%) | 1 (1.4%) | |
| Severe | 0 (0%) | 0 (0.0%) | 0 (0.0%) | |
| *Renal and urinary disorders* | | | | NA |
| Mild | 1 (2.3%) | 0 (0.0%) | 1 (1.4%) | |
| Moderate | 0 (0%) | 0 (0.0%) | 0 (0.0%) | |
| Severe | 0 (0%) | 0 (0.0%) | 0 (0.0%) | |
| *Respiratory disorders* | | | | NA |
| Mild | 0 (0.0%) | 0 (0.0%) | 0 (0.0%) | |
| Moderate | 0 (0.0%) | 1 (3.6%) | 1 (1.4%) | |
| Severe | 0 (0.0%) | 0 (0.0%) | 0 (0.0%) | |
| *Skin disorders* | | | | NA |
| Mild | 0 (0%) | 0 (0.0%) | 0 (0.0%) | |
| Moderate | 1 (2.3%) | 0 (0.0%) | 1 (1.4%) | |
| Severe | 0 (0%) | 0 (0.0%) | 0 (0.0%) | |
| *Vascular disorders* | | | | NA |
| Mild | 0 (0.0%) | 0 (0.0%) | 0 (0.0%) | |

**Table 2 (continued)**

| Body system preferred term severity | Tofacitinib adverse events = 43 | Placebo adverse events = 28 | Total adverse events = 71 | P value |
|---|---|---|---|---|
| Moderate | 0 (0.0%) | 0 (0.0%) | 0 (0.0%) | |
| Severe | 0 (0.0%) | 1 (3.6%) | 1 (1.4%) | |

Fisher's exact test was used to calculate the *P* values and compare the incidents of the adverse events between tofacitinib and placebo. Two-tailed tests were used where appropriate. No adjustments were made for multiple comparisons.

decreased compared to baseline in *STAT4* risk allele-positive subjects that received tofacitinib ($P = 0.037$) (Fig. 2d). In contrast, subjects receiving placebo or those subjects receiving tofacitinib that were *STAT4* risk allele-negative did not display decreases in circulating NET complexes (Fig. 2d). Overall, these results indicated that *STAT4* risk allele-positive subjects displayed higher levels of circulating NETs and that tofacitinib modified features of SLE-associated neutrophil dysregulation in subjects positive for this genetic risk haplotype[7]. However, even though these differences were significant but there was an overlap between the groups and there was also a lower NET expression in placebo subjects at the baseline.

**Improvements in cardiometabolic parameters and vascular function.** Since tofacitinib improved type I IFN signaling and neutrophil abnormalities associated with premature CVD, we determined if tofacitinib treatment affected plasma lipoprotein levels and function. There was a significant increase in HDL-C in all subjects in the tofacitinib group at the completion of the on-study drug phase (day 56) compared to baseline ($P = 0.006$) regardless of their *STAT4* risk allele status (Fig. 3a and Supplementary Fig. 5b). Low-density lipoprotein levels and triglycerides were not affected (Supplemental Fig. 5a–d). Tofacitinib also significantly increased levels of large HDL particles, whereas other lipoprotein subfractions did not show significant changes (Table 4). At day 56, a significant increase in Lecithin–cholesterol acyltransferase (LCAT) concentration was detected in SLE subjects on tofacitinib that were *STAT4* risk allele-positive ($P = 0.024$, 95% CI: 1.1–26.5) compared to the *STAT4* risk allele-positive group on placebo or in subjects negative for this risk allele on tofacitinib ($P = 0.04$, 95% CI: −18.2, 10.2) (Fig. 3b). In addition, at day 56, there was a statistically significant increase in cholesterol efflux capacity in SLE subjects treated with tofacitinib ($P = 0.002$, 95% CI: 0.04–0.16) and a nonsignificant trend when compared to placebo ($P = 0.08$, 95% CI: −0.01, 0.24) (Supplementary Fig. 5e). This effect was not dependent on the subject's *STAT4* risk allele status. No statistically significant differences in insulin resistance, as measured by HOMA-IR, were detected between placebo and tofacitinib-treated groups ($P = 0.51$, 95% CI: −0.93, 0.48) (Table 3). Compared to the tofacitinib group, the placebo group had lower cholesterol and triglyceride values at the baseline, but these differences were not statistically significant and remained essentially unchanged throughout the study (Table 3).

Enhanced arterial stiffness and impairments in endothelium-dependent vasorelaxation have been described in association with vasculopathy and subclinical atherosclerosis in SLE[20,21]. Consistent with previous studies, the baseline values for various measures of arterial stiffness and impairments in endothelial dysfunction in SLE subjects were higher than the reference values for age-matched controls (Table 1)[22,23]. Patients in the tofacitinib

**Table 3 Longitudinal follow-up of laboratory values during the trial.**

| Outcome variable | Tofacitinib (n = 20) | | | Placebo (n = 10) | | | P value baseline | P value D56, difference in change scores (95% CI) | P value D84, difference in change scores (95% CI) |
|---|---|---|---|---|---|---|---|---|---|
| | Day 1 mean ± SD | Day 56 mean ± SD | Day 84 mean ± SD | Day 1 mean ± SD | Day 56 mean ± SD | Day 84 mean ± SD | | | |
| WBC, K/mcL | 5.3 ± 1.6 | 5.2 ± 1.7 | 4.8 ± 1.5 | 4.8 ± 1.1 | 5.3 ± 1.6 | 4.9 ± 1.1 | 0.32 | 0.13, −0.63 (−0.63, −1.46) | 0.19, −0.52 (−1.35, 0.3) |
| Absolute neutrophil counts, K/uL | 3 ± 1.3 | 2.9 ± 1.3 | 2.7 ± 1.1 | 2.7 ± 0.9 | 3.2 ± 1.3 | 2.8 ± 0.7 | 0.48 | 0.14, −0.58 (−0.58, −1.35) | 0.5, −0.21 (−0.85, 0.43) |
| Hemoglobin, g/dL | 12.8 ± 1.2 | 12.6 ± 1.2 | 12.4 ± 1.4 | 12.9 ± 1.5 | 13 ± 1.9 | 12.8 ± 2 | 0.8 | 0.24, −0.33 (−0.33, −0.88) | 0.5, −0.33 (−0.81, 0.40) |
| Platelet, K/mcL | 224 ± 72.5 | 221.6 ± 74.9 | 231.5 ± 68.4 | 213.4 ± 57.7 | 225.9 ± 67.1 | 213.7 ± 54.2 | 0.69 | 0.13, −15.36 (−15.36, −35.79) | 0.54, 5.3 (−12.33, 22.93) |
| ESR, mm/h | 22 ± 19.2 | 19.4 ± 17.2 | 22.6 ± 17.6 | 25.9 ± 24.3 | 25.7 ± 26.2 | 25.7 ± 25.6 | 0.63 | 0.2, −3.74 (−3.74, −9.63) | 0.77, 0.64 (−3.82, 5.10) |
| CRP, mg/L* | 4.6 ± 7.7 | 2.3 ± 3.2 | 3.2 ± 3.4 | 2.7 ± 4 | 3.3 ± 5.1 | 4 ± 8.2 | 0.36 | 0.24, 0.59 (0.23, 1.46) | 0.84, 0.95 (0.56, 1.60) |
| C3, mg/dL | 106.6 ± 26.1 | 100.4 ± 21.6 | 107.4 ± 24.1 | 108.8 ± 28.6 | 106.8 ± 29.7 | 108.5 ± 32.2 | 0.84 | 0.34, −4.48 (−4.48, −13.96) | 0.85, 0.82 (−7.76, 9.39) |
| C4, mg/dL | 22 ± 10.7 | 19.8 ± 10 | 21.1 ± 9.8 | 19.4 ± 8.2 | 18.8 ± 7.4 | 19 ± 8.5 | 0.51 | 0.32, −1.17 (−1.17, −3.53) | 0.88, −013 (−1.95, 1.69) |
| Anti dsDNA, IU/mL** | 98.6 ± 81.7 | 127.4 ± 85.9 | 157.2 ± 115.7 | 55.0 ± 21.7 | 61.3 ± 21.6 | 99.7 ± 70.7 | 0.41 | 0.14, 40.08 (−20.71, 100.87) | 0.96, 3.46 (−163.58, 170.50) |
| AST, u/L | 20.6 ± 5.3 | 22.4 ± 6 | 21.5 ± 5.6 | 21.7 ± 4.5 | 22.2 ± 9.2 | 35.1 ± 45.7 | 0.56 | 0.8, 0.71 (0.71, −5.14) | 0.19, −13.8 (−34.91, 7.31) |
| ALT, u/L | 17.7 ± 9 | 20.1 ± 11.5 | 19.6 ± 10.6 | 19.7 ± 8.6 | 20.3 ± 14.3 | 23.1 ± 13 | 0.56 | 0.6, 1.58 (1.58, −4.55) | 0.52, −2.15 (−8.86, 4.56) |
| Cholesterol, mg/dL | 170.5 ± 31.3 | 184.9 ± 45.3 | 170.6 ± 33.7 | 148.4 ± 23.2 | 147.2 ± 26.1 | 149.6 ± 31.6 | 0.06 | 0.25, 9.55 (−6.96, 26.06) | 0.91, −0.77 (−15.14, 13.61) |
| Triglycerides, mg/dL* | 108.7 ± 46.6 | 115.8 ± 107.7 | 107 ± 65.7 | 79.8 ± 29.9 | 84.5 ± 43.3 | 90.7 ± 44.2 | 0.09 | 0.97, 0.99 (0.68, 1.46) | 0.78, 0.95 (0.68, 134) |
| HDL, mg/dL | 56.5 ± 17.5 | 64.2 ± 18.8 | 56.5 ± 19.7 | 52.2 ± 16.4 | 51.4 ± 15.6 | 51.5 ± 19.3 | 0.53 | 0.0006, 8.72 (4.12, 13.32) | 0.92, 0.22 (−3.94, 4.38) |
| LDL, mg/dL | 92.2 ± 30 | 94.2 ± 31.6 | 92.8 ± 27.8 | 80.2 ± 23.4 | 79 ± 25.2 | 80.2 ± 24.2 | 0.28 | 0.29, 6.68 (−5.98, 19.33) | 0.61, 2.56 (−7.61, 12.72) |
| HDL particle number, mcmol/L | 32.1 ± 6.5 | 34.7 ± 6.1 | 31.2 ± 6.4 | 28.9 ± 5.6 | 28.4 ± 5.6 | 28.1 ± 7 | 0.2 | 0.0008, 3.46 (1.58, 5.33) | 0.98, 0.02 (−2.09, 2.14) |
| HDL size, nm | 9.4 ± 0.5 | 9.6 ± 0.6 | 9.5 ± 0.5 | 9.4 ± 0.6 | 9.5 ± 0.5 | 9.5 ± 0.5 | 0.86 | 0.85, 0.02 (−0.23, 0.28) | 0.37, −0.09 (−0.31, 0.12) |
| LDL particle number, nmol/L | 1024.8 ± 396.9 | 1055.5 ± 556.5 | 1053.3 ± 433 | 898.4 ± 286.8 | 869 ± 309.7 | 931.1 ± 372.9 | 0.23 | 0.71, 25.76 (−172.31, 223.82) | 0.95, −4.66 (−168.02, 158.71) |
| LDL size, nm | 20.7 ± 0.5 | 20.9 ± 0.7 | 20.7 ± 0.7 | 20.7 ± 0.6 | 20.7 ± 0.5 | 20.5 ± 0.7 | 0.89 | 0.28, 0.2 (−0.17, 0.57) | 0.29, 0.17 (−0.16, 0.50) |
| VLDL particle number, nmol/L | 43.9 ± 20 | 50.5 ± 40.1 | 44.1 ± 39.5 | 31.4 ± 18.3 | 34.7 ± 26.5 | 39.2 ± 25.4 | 0.11 | 0.58, 8.14 (−21.7, 37.98) | 0.83, −3.1 (−31.87, 25.68) |
| VLDL size, nm* | 48.9 ± 4.8 | 47.8 ± 5.6 | 49.1 ± 7.6 | 46 ± 7.1 | 47.3 ± 4.5 | 45.6 ± 4.8 | 0.15 | 0.70, 0.98 (0.9, 1.07) | 0.45, 1.04 (0.93, 1.16) |
| Glucose, mmol/l | 5 ± 0.7 | 4.9 ± 0.7 | 5.2 ± 1 | 4.9 ± 0.4 | 4.9 ± 0.6 | 5 ± 1.1 | 0.57 | 0.18, −0.16 (−0.16, −0.4) | 0.97, −0.007 (−0.57, 0.55) |
| Insulin, Pmol/L | 98.7 ± 42 | 88.3 ± 35.9 | 96.4 ± 38.7 | 143.1 ± 95.1 | 131.3 ± 104.4 | 204.3 ± 246.2 | 0.19 | 0.54, −12.19 (−52.40, 28.01) | 0.45, −29.12 (−106.62, 48.39) |
| HOMA2 IR | 1.8 ± 0.8 | 1.6 ± 0.7 | 1.8 ± 0.7 | 2.6 ± 1.7 | 2.4 ± 1.9 | 3.6 ± 4.2 | 0.21 | 0.51, −0.23 (−0.93, 0.48) | 0.38, −0.56 (−1.85, 0.74) |

WBC white blood cell, ANC absolute neutrophil count, ESR erythrocyte sedimentation rate, CRP C-reactive protein, C3 complement component 3, C4 complement component 4, anti-ds-DNA anti-double-stranded DNA antibody, AST aspartate aminotransferase, ALT alanine aminotransferase, HDL high-density lipoprotein, LDL low-density lipoprotein, VLDL very-low-density lipoprotein, mcmol/L micromoles per liter, nm nanometer, Pmol/L picomole per liter, HOMA2 IR Homeostatic Model Assessment Index for Insulin Resistance.
Baseline P values were calculated using the two-sample t.tests. P values and the 95% confidence intervals for treatment group difference in change score from baseline to day 56 were calculated using either (1) linear mixed-effects models if there were multiple assessment time points between baseline and day 56 or (2) analysis of covariance models (ANCOVA) if the measures were collected only at baseline and day 56 during the treatment phase. For day 84, ANCOVA was used to compare the change scores between treatment groups. The STAT4 allele status and the baseline values were included as covariates in the linear mixed-effects models and ANCOVA.
*P values were calculated for the log-transformed data in order to meet the model normality assumptions. The estimates and the 95% confidence bounds were exponentiated back.
**Data are based on 5/20 subjects in the tofacitinib group and 3/10 subjects in the placebo group who had positive anti-dsDNA results. Wilcoxon ranked-sum test was used to calculate the P values.
Two-tailed tests were used where appropriate.
No adjustments were made for multiple comparisons.

**Table 4 Disease activity and patient-reported outcome measures.**

| Outcome variable | Tofacitinib (n = 20) | | | Placebo (n = 10) | | | | Difference in change (tofacitinib vs. placebo) | |
|---|---|---|---|---|---|---|---|---|---|
| | Day 1 mean ± SD | Day 56 mean ± SD | Day 84 mean ± SD | Day 1 mean ± SD | Day 56 mean ± SD | Day 84 mean ± SD | Day 1 P value | Day 56* P value difference (95% CI) | Day 84** P value difference (95% CI) |
| SF-36 total | 110.3 ± 9.6 | 113.3 ± 9.5 | 111.9 ± 9.3 | 107.6 ± 8.4 | 111.5 ± 8.1 | 108 ± 9.8 | 0.46 | 0.91, −0.20 (−3.69, 3.29) | 0.31, 2.92 (−2.84, 8.69) |
| MD-Fatigue average | 1.6 ± 1.7 | 1.9 ± 2.2 | 2.1 ± 2.2 | 4.4 ± 2.7 | 2.2 ± 1.6 | 2.3 ± 2.2 | 0.001 | 0.016, 1.47 (0.30, 2.65) | 0.04, 1.71 (0.04, 3.36) |
| SLEDAI-2K | 5.1 ± 2.2 | 4.2 ± 2.3 | 3.9 ± 1.9 | 5.5 ± 3.7 | 4.5 ± 2.4 | 4.9 ± 2.9 | 0.68 | 0.95, 0.04 (−1.04, 1.11) | 0.25, −0.72 (−1.98, 0.53) |
| PGA | 0.8 ± 0.8 | 0.8 ± 0.9 | 0.9 ± 0.7 | 1.2 ± 0.9 | 0.9 ± 0.8 | 0.8 ± 0.8 | 0.21 | 0.43, 0.18 (−0.28, 0.64) | 0.38, 0.23 (−0.30, 0.76) |
| DAS-28-ESR | 2.7 ± 1.3 | 2.5 ± 1.2 | 2.5 ± 0.9 | 3.2 ± 2 | 2.9 ± 1.6 | 2.8 ± 1.3 | 0.44 | 0.73, −0.09 (−0.61, 0.43) | 0.81, 0.05 (−0.40, 0.51) |
| CLASI total activity | 2.6 ± 1.9 | 1.9 ± 1.7 | 2.1 ± 1.8 | 2.4 ± 1.7 | 2.1 ± 1.7 | 2.2 ± 1.5 | 0.83 | 0.35, −0.31 (−1.03, 0.42) | 0.54, −0.26 (−1.12, 0.60) |
| CLASI total damage*** | 1 ± 2.1 | 1.1 ± 2.2 | 1.1 ± 2.2 | 1 ± 2.2 | 0.9 ± 2.2 | 0.9 ± 2.2 | 1 | 0.46 | |
| BILAG 2004† | 7.6 ± 4.6 | 6.2 ± 4.8 | 5.5 ± 3.9 | 9.3 ± 4.3 | 8.1 ± 5.5 | 7.9 ± 4.4 | 0.32 | 0.36, 1.56 (−1.86, 4.98) | 0.16, −2.04 (−4.96, 0.89) |

SF 36 Short Form Health Survey, MD-Fatigue Multidimensional Assessment of Fatigue Questionnaire, SLEDAI 2K Systemic Lupus Erythematosus Disease Activity Index 2000, PGA Physician Global Assessment, DAS 28-ESR Disease Activity Score of the 28 joints with erythrocyte sedimentation rate, CLASI Cutaneous Lupus Erythematosus Disease Area and Severity Index, BILAG 2004 British Isles Lupus Assessment Group Disease Activity Index.
*To calculate the P value and the estimated difference (95% confidence interval) in change from baseline (tofacitinib vs. placebo), the linear mixed-effects models were used to include all the other time points between baseline and day 56.
**To calculate the P value and the estimated difference (95% confidence interval) in change from baseline (tofacitinib vs. placebo), the analysis of covariance (ANCOVA) test was used to analyze data on day 84 only.
***Due to the violation of the normality assumption, the Wilcoxon ranked-sum test was used to assess the baseline data and to compare the change scores for day 56 and day 84 separately. The 95% confidence intervals were not provided.
†BILAG 2004 numerical scoring method: A = 12, B = 8, C = 1, D = 0, and E = 0.
Two-tailed tests were used where appropriate.
No adjustments were made for multiple comparisons.

arm had clinically significant decreases in arterial stiffness (as measured by CAVI and PWV) and improvement in endothelium-dependent vasorelaxation (as measured by Endo-PAT RHI). Specifically, patients on tofacitinib had decreases in CAVI by mean ± standard error of the mean of −0.28 ± 0.14 at day 56 and −0.14 ± 0.19 at day 84 (Fig. 4a). PWV decreased only in the *STAT4* risk allele-positive subjects on tofacitinib by −0.42 m/s ± 0.58 at day 56, and −1.41 m/s ± 0.92 at day 84, respectively, from baseline (Fig. 4b). Similarly, there was an improvement in RHI, in *STAT4* risk allele-positive subjects on tofacitinib, by 0.18 ± 0.17 at day 56, and 0.23 + 0.09 at day 84 (Fig. 4c). In contrast, no improvements were observed in the placebo group. Overall, these results indicated that SLE subjects on short-term tofacitinib treatment exhibited improvements in cardiometabolic and vascular parameters previously associated with CV risk in SLE patients and in the general population. Importantly, some of these protective effects were more apparent in subjects bearing the *STAT4* risk allele.

## Discussion

In this study, short-term use of tofacitinib in subjects with mild-to-moderate SLE was overall safe, well-tolerated, with no unexpected AEs, thromboembolic events or opportunistic infections. Our exploratory analyses showed that tofacitinib led to significant positive modulation of cardiometabolic and immunologic parameters previously linked to increased coronary atherosclerotic plaque, vascular inflammation, and abnormalities in blood vessel function in lupus and in the general population[24,25]. Premature atherosclerosis and vasculopathy leading to CV events significantly contribute to enhanced morbidity and mortality in SLE, a phenomenon not explained by traditional CV risk factors[6,24]. The observation that jakinibs may have a possible role in modulating SLE vasculopathy and potentially mitigating CV risk could have important implications in this patient population.

Previous work from our group and others has implicated a pathophysiologic alliance between type I IFNs, LDGs, and enhanced NET formation as a mechanism that promotes premature atherosclerosis and vasculopathy in SLE[7,26–29]. Tofacitinib significantly decreased the type I IFN gene signature in circulating immune cells and selectively modulated the numbers of circulating LDGs, but not total neutrophils, in association with significant decreases in levels of circulating NETs. The mechanism by which tofacitinib led to decreases in LDGs remains to be further characterized and may be related to inhibition of cytokine-specific effects on these cells in the bone marrow or tissues, perhaps promoting their death through apoptosis. Of interest, we found that individuals carrying the *STAT4* risk allele had higher levels of circulating NETs, suggesting that certain polymorphisms associated with an enhanced risk for auto-immunity may promote the enhanced NET formation, a phenomenon recently reported for other autoimmunity risk SNPs[30,31].

While the effect on type I IFN was expected, it remains unclear why LDGs were preferentially targeted by tofacitinib and why the response to the drug concerning NET inhibition was enhanced in the presence of the *STAT4* risk allele. One scenario is that since type I IFNs activate *STAT4*, subjects bearing the *STAT4* risk allele have increased priming of neutrophils resulting in increased NET formation. Our current data are consistent with this view and it appears that tofacitinib may have preferentially blocked this priming in subjects with the risk allele[32]. While tofacitinib affected type I IFN responses and markers of neutrophil dysregulation, the effects observed on other immune cell types or clinical activity in subjects with mild-to-moderate SLE were not significant. The exact role of type I IFN in the pathogenesis of SLE

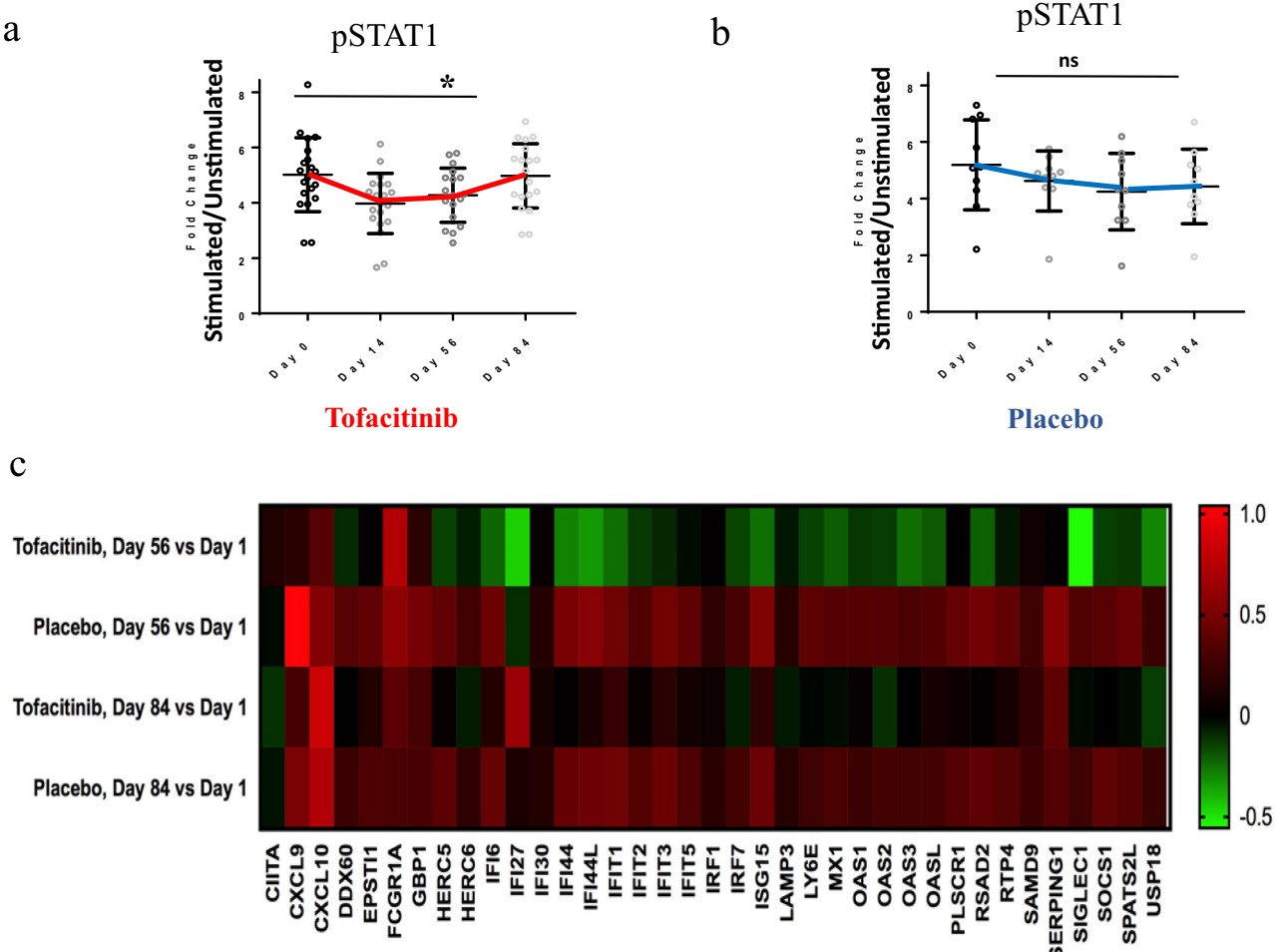

**Fig. 1 Tofacitinib inhibits STAT1 phosphorylation in CD4 + T cells. a** Significant inhibition of pSTAT1 in patients on tofacitinib at day 56 (P = 0.023) with the return to baseline at day 84 n = 20 biologically independent samples. Data are presented as mean values +/− SEM. A mixed linear model for repeated measures was used. **b** No significant change noted in patients on placebo. n = 10 biologically independent samples. Data are presented as mean values +/− SEM. A mixed linear model for repeated measures was used. **c** Tofacitinib decreases the type I IFN gene signature, gene expression in peripheral blood by Nanostring: Heatmap showing interferon-stimulated gene expression in peripheral blood by nanostring: Log2 mean fold change in expression of interferon-stimulated genes from baseline to day 56 and 84 in subjects on tofacitinib (n = 20 biologically independent samples) and placebo (n = 10 biologically independent samples). Source data are provided as a Source Data file. Results shown as fold change in stimulated vs. unstimulated cell population. A mixed linear model for repeated measures was used. Unpaired two-tailed, t test were used where appropriate. No adjustments were made for multiple comparisons.

is still being defined and abrogating this pathway may not lead to decreases in disease manifestations, as evidence from recent clinical trials using interferon receptor blocker Anifrolumab and the plasmacytoid dendritic cells specific receptor antibody BIIB059[33,34]. Future studies on the effects of tofacitinib in larger patient groups, stratified by genetic risk and greater active disease, will be needed to assess alteration of both innate and adaptive immune parameters, and clinical disease activity.

SLE HDL has been described to be small particle size and with decreased cholesterol efflux capacity and LCAT activity driven at least in part by oxidation through enhanced NET formation[7,35–37]. Herein, we show that short-term use of tofacitinib in SLE led to significant improvements in the levels, size, and function of HDL. Importantly, tofacitinib led to improvements in arterial compliance and endothelial function, with the reversal of SLE-associated accelerated vascular aging. The improvement in aortic compliance as measured by PWV persisted during the washout phase possibly due to modulatory effects of tofacitinib on endothelial cells[38]. Even though these changes did not reach statistical significance (probably due to the small sample size) yet the extent of change in a short duration is impressive and requires further exploring. The mechanism of this finding of short-term tofacitinib exposure leading to improvements in arterial compliance and endothelial function remains speculative. Higher prevalence of the *STAT4* risk allele has been associated with cardiovascular risk in SLE patients and atherosclerosis animal models[16,39]. In this study, HDL function (as measured by LCAT activity), arterial stiffness, and endothelium-dependent vasorelaxation improved in individuals on tofacitinib that were positive for *STAT4* risk allele.

Future clinical trials will need to establish whether the positive effects on vascular function by jakinibs are disease-specific or are generalizable to other autoimmune disorders and, perhaps, even the general population. Indeed, addressing CV risk in a non-autoimmune disease population by using anti-inflammatory medications (canakinumab) has been reported to be successful[25].

There was a nonsignificant trend toward lower absolute neutrophil counts in subjects on tofacitinib. As expected, there were

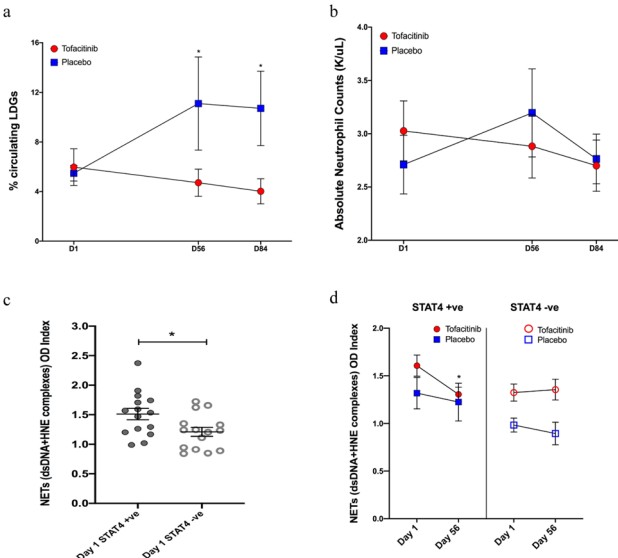

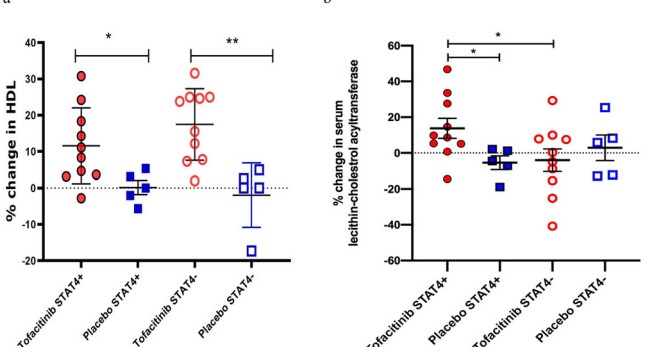

**Fig. 2 Circulating LDGs and NETs are modulated by tofacitinib. a** Results represent changes in percentage of circulating LDGs: Subjects treated with tofacitinib vs. placebo. Significant decrease in LDGs in tofacitinib group $P = 0.048$ at day 56 and $P = 0.014$ at day 84 using unpaired $t$ test. **b** Changes in absolute neutrophil counts. **c** Circulating NET levels at baseline: Individuals positive for *STAT4* risk allele (each subject represented by closed circles) have enhanced circulating NET levels (assessed by human neutrophil elastase (HNE)-dsDNA complexes) then subjects who are *STAT4* risk allele negative (each subject represented by open circles) $P = 0.02$ using unpaired $t$ test. **d** Changes in circulating NET levels during the study: Patients who are *STAT4* risk allele positive and receive tofacitinib display significant decrease in circulating NET complexes $P = 0.037$ using unpaired $t$ test. Source data are provided as a Source Data file. All data represent mean ± SEM, *$P < 0.05$, and are based on tofacitinib $n = 20$, placebo $n = 10$. Unpaired two-tailed, $t$ test were used for all results. No adjustments were made for multiple comparisons.

more mild and moderate infections (mostly upper respiratory tract infections) in the tofacitinib group as compared to placebo. Some of the limitations of the current study include its short duration and a small number of subjects with mild-to-moderate disease. Being an early phase exploratory study, adjustments were not made for multiple comparisons. In addition, there were differences in the baseline between the groups and changes observed in the placebo group which may be partially responsible for some of the results observed in the secondary outcomes of this study. No concomitant immunosuppressive agents were allowed and therefore the safety profile in presence of additional drugs remains to be characterized. Nevertheless, using a drug that has a good safety profile and promotes beneficial effects on systemic inflammation, disease manifestations and CV risk prevention would be a highly desired target for SLE and other autoimmune diseases.

In summary, our study used a personalized approach in the context of SLE by stratifying patients according to *STAT4* risk allele. Results from the current trial suggest the effect of JAK inhibition was more robust in subjects with *STAT4* risk allele, which is associated with a more severe SLE, and an increased risk of CV events[15,16]. The JAK-STAT pathway is involved in the intracellular signaling of multiple cytokines; therefore, additional mechanistic studies are needed to better characterize the pathways responsible for findings in this study. Larger studies with longer follow-up can better assess if this personalized medicine approach of stratification by genetic risk allele leads to improved morbidity and mortality.

**Fig. 3 Tofacitinib modulates HDL levels and function in SLE. a** Percent change in serum HDL-C at day 56 compared to day 1 based on *STAT4* risk allele status (each circle and square represent individual subject): Results represent *$P = 0.037$ for the difference between *STAT4* risk allele-positive subjects on tofacitinib vs placebo and **$P = 0.002$ *STAT4* risk allele-negative subjects on tofacitinib vs placebo. Unpaired $t$ test was used. **b** Percent change in Lecithin–cholesterol acyltransferase (LCAT) concentration at day 56 compared to day 1 based on *STAT4* risk allele status (each circle and square represent individual subject): Results represent *$P = 0.033$ for the difference between *STAT4* risk allele-positive subjects on tofacitinib vs placebo and *$P = 0.044$ for the difference between *STAT4* risk allele-positive subjects on tofacitinib vs *STAT4* risk allele-negative subjects on tofacitinib. Kruskal–Wallis test (unpaired, nonparametric) was used. Source data are provided as a Source Data file. All results represent mean ± SEM, *$P < 0.05$ **$P < 0.01$, and are based on tofacitinib $n = 20$, placebo $n = 10$. Two-tailed tests were used where appropriate. No adjustments were made for multiple comparisons.

## Methods

**Study design and subjects**. The study was approved by the National Institutes of Health (NIH) IRB (ClinicalTrials.gov NCT02535689). The study design and conduct complied with all relevant regulations regarding the use of human study participants and was conducted in accordance with the criteria set by the Declaration of Helsinki as authorized by the NIH Office of Human Subject Research. After written informed consent and determination of eligibility, subjects were randomized to tofacitinib (5 mg twice daily) or placebo in a 2:1 block. Thirty adult SLE subjects that met American College of Rheumatology (ACR) Revised Criteria for the Classification of SLE and had mild-to-severe disease activity (Systemic Lupus Erythematosus Disease Activity Index 2000 (SLEDAI 2 K) score between 2 and 14) were enrolled in an outpatient clinical research setting[40]. In each group, half of the subjects were homozygous or heterozygous for the *STAT4* risk allele. Subjects took tofacitinib or placebo for 8 weeks and were followed for 4 additional weeks afterward (Supplementary Fig. 1). Eligible subjects were on stable doses of antimalarials (for 12 weeks prior to the screening visit) and/or glucocorticoids (for 4 weeks prior to the screening visit) (prednisone or equivalent <20 mg/day) but no immunosuppressants were allowed. The SLE disease activity was determined using SLEDAI 2 K, BILAG 2004, Disease Activity Score 28-Erythrocyte Sedimentation Rate (DAS-28-ESR), Physician Global Assessment (PGA) (Likert scale 0–3), and patient-reported outcomes (SF-36, Multidimensional Assessment of Fatigue questionnaire)[41–46]. The rates of adverse events (AEs, as defined by the National Cancer Institute (NCI), Common Terminology Criteria for Adverse Events (CTCAE), Version 4.0) were recorded weekly. The primary outcome of the study was defined as comparing rates of adverse events and rates of SLE disease flares between the tofacitinib group and the placebo group. The SLE disease flare was defined as an increase in SLEDAI 2 K score of ≥3 or an increase in PGA > 1. The secondary outcomes included assessment of clinical response, effects on quality of life measures, and several exploratory mechanistic studies to evaluate the effect of the drug on immune dysregulation and cardiometabolic signatures associated with the development of premature cardiovascular disease (the original protocol, final protocol, and summary of changes are provided as part of the Supplementary files).

**Immunologic and hematologic parameters**. Each subject had *STAT4* risk allele genotyping prior to screening visit using commercially available TaqMan® Single Nucleotide Polymorphism Genotyping. Detailed reagents and methods for PBMC isolation, stimulation, and the fluorescent barcoding method used for the staining and multiplexed phosphoflow analysis are described in Supplementary Methods and elsewhere[47]. Pathway analysis and gene annotation were completed using Toppgene (toppgene.cchmc.org). Results were confirmed using NanoString

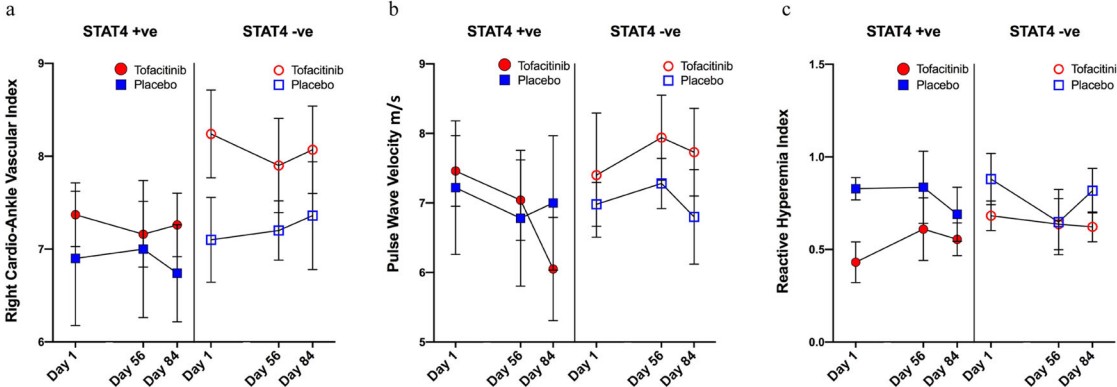

**Fig. 4 Tofacitinib improves arterial stiffness and endothelial dysfunction in association with *STAT4* risk allele status. a** Right cardioankle vascular index (CAVI): Results represent changes in CAVI in subjects on tofacitinib vs placebo in *STAT4* risk allele-positive and -negative subjects during the trial. **b** Pulse wave velocity (PWV): Results represent changes in PWV in subjects on tofacitinib vs placebo in *STAT4* risk allele-positive and -negative subjects during the trial. **c** Reactive hyperemia index (RHI): Results represent changes in RHI in subjects on tofacitinib vs placebo in *STAT4* risk allele-positive and -negative subjects during the trial. Source data are provided as a Source Data file. All results represent mean ± SEM and are based on tofacitinib $n = 20$, placebo $n = 10$. The paired *t* test, Mann–Whitney *U,* or ANOVA were used for comparison where appropriate based on normality of distribution. Two-tailed tests were used where appropriate. No adjustments were made for multiple comparisons.

(see Supplementary Methods). LDGs were classified as CD10[+], CD15[+], CD14[lo48]. Human neutrophil elastase (HNE)-DNA NET complexes were measured in plasma, as described[49]. Details are provided in Supplementary Methods.

**Measurements of cardiovascular risk factors and vascular function**. The homeostasis model assessment of insulin resistance (HOMA-IR) index = fasting glucose (mmol/l) × fasting insulin (μU/ml)/22.5) was used to estimate insulin resistance[50]. Overnight fasting lipid profiles were performed at the NIH Clinical Center Central Laboratories. Lipoprotein particle concentration and diameters were measured using an automated Nuclear Magnetic Resonance Spectroscopy (NMR). High-density lipoprotein (HDL) cholesterol efflux capacity was measured using published methods and the macrophage cell line J774[51]. Human serum lecithin–cholesterol acyltransferase (LCAT) concentration was quantified by ELISA (BioVendor; Ashville, NC). Noninvasive vascular function studies included the cardioankle vascular index (CAVI), peripheral arterial tonometry (PAT; reactive hyperemia index (RHI), and pulse wave velocity (PWV)[7]. Details are provided in the Supplementary Methods.

**Statistical analysis**. The sample size chosen was based on what is commonly being used in similar studies and consistent with our experiences in early phase safety studies[52,53]. No formal power calculations were performed. Data from all randomized subjects were included in the analysis. The adverse events were summarized with frequency counts and percentages for each treatment group. To evaluate the treatment effect on the change from baseline to the end of the treatment period (day 56), a linear mixed-effects model approach was utilized to fit the longitudinal data on continuous safety parameters and clinical outcomes. The model included the baseline value, the *STAT4* risk allele status, the treatment group, the categorical time point, and the treatment by time interaction as fixed effects. An unstructured variance-covariance matrix was used to account for the correlations among repeated measures. For variables assessed only at baseline and day 56 during the treatment period, the analysis of covariance (ANOCA) models was used including baseline, treatment, and the *STAT4* risk allele status as the covariates. In addition, paired *t* test, Mann–Whitney *U,* or ANOVA were used for comparison where appropriate based on normality of distribution. Change scores from baseline to day 84 were analyzed separately using ANCOVA models. No adjustments for multiple comparisons were made to the *P* values due to the exploratory nature of the analysis. The first patient was enrolled on July 28, 2016 and the last patient was enrolled on September 18, 2017. All statistical analyses were performed with SAS software (version 9.4).

**RNA sequencing analysis**. Gene expression values were calculated with Partek Genomics Suite 6.6, which was also used for the principal components analysis (PCA) and one-way ANOVA. The ANOVA was performed on log2 transformed RPKM with a 0,1 offset.

**NanoString data analysis**. Assessment of the quality of the runs was done (Supplementary Methods). Data were combined, normalized, and analyzed in Microsoft Excel (Microsoft Corporation; Redmond, WA). JMP version 14 was used for further statistical analysis and plotting (SAS Corporation; Cary, NC). Synthetic DNA oligonucleotides of each of the 37 ISGs and 4 housekeeping genes were used as a calibration standard to check run and reagent lot consistency.

**Reporting summary**. Further information on research design is available in the Nature Research Reporting Summary linked to this article.

## Data availability

RNA-Seq results have been deposited in the Gene Expression Omnibus (GEO) database; Geo Series Entry GSE139940. The study protocol and statistical analysis plan are submitted as part of the Supplementary File. In addition, the source data for fluorescent cell barcoding is provided as a separate file and can be accessed as referenced[54]. The RNA-Seq and fluorescent cell barcoding data are available without any material transfer and sharing agreement. Any other data are available after executing material transfer and sharing agreement as applicable under laws of the US Government. Please contact Sarfaraz A. Hasni, hasnisa@mail.nih.gov for any data-sharing requests. Source data are provided with this paper.

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

## Acknowledgements

This research was supported by the Intramural Research Program of the National Institute of Arthritis and Musculoskeletal and Skin Diseases of the National Institutes of Health. Pfizer Inc. provided study drug and placebo for this trial. The trial was funded by the Intramural Research Program, NIAMS/NIH. The drug and placebo were provided by Pfizer Inc. through an investigator-initiated research grant. The role of Pfizer was limited to providing study drug and placebo, they had no role in design or conduct of the study, reporting of results, or manuscript preparation.

## Author contributions

S.A.H.: study concept, design, protocol development, conduct, analyses, and manuscript draft. S.G.: study design, protocol development, conduct, analyses, and manuscript review. M.D.: study conduct and manuscript review. E.P.: study conduct and manuscript review. Y.T-O.: study conduct and manuscript review. P.M.C.: performed experimental studies and manuscript review. X.W.: performed experimental studies and manuscript review. M. N.: performed experimental studies, results analyses, and manuscript review. M.P.P.: performed experimental studies and manuscript review. R.R.G.: performed experimental studies and manuscript review. X.L.: performed statistical analyses. A.J.B.: study conduct and manuscript review. I.O.-N.: study conduct. Z.M.: data management and manuscript review. Y.S.: study conduct. D.T.: study conduct and manuscript review. J.C.: performed experimental studies. A.B.: performed experimental studies and manuscript review. R.A.: performed experimental studies and manuscript review. F.C.: performed experimental studies and manuscript review. Y.K.: performed experimental studies and manuscript review. A.L.B.: performed experimental studies and manuscript review. H.Z.: performed experimental studies and manuscript review. R.S.: performed experimental studies and manuscript review. K.S.: performed experimental studies and manuscript review. W. L.T.: performed experimental studies. L.V.: performed experimental studies. N.G.: performed experimental studies. V.G.: performed experimental studies. S.L.: performed experimental studies, analyses, and manuscript review. S.R.B.: performed experimental studies, analyses, and manuscript review. M.M.: study design and manuscript review. P.G.: study design, performed experimental studies, and manuscript review. N.N.M.: performed experimental studies and manuscript review. A.T.R.: performed experimental studies, analyses, and manuscript review. B.D.: study concept, design, and manuscript review. J.J.O.: study concept, design, and manuscript review. M.G.: study conduct, analyses, and manuscript review. M.J.K.: study concept, design, conduct, analyses, manuscript draft, and review.

## Funding

## Competing interests

The NIH and J.J.O.S. have a patent related to JAK inhibitors and receive royalties. The NIH and J.J.O.S. have had a collaborative agreement and development award (CRADA) with Pfizer that pertains to JAK inhibition and tofacitinib. The NIH and J.J.O.S. have an ongoing CRADA for new JAK inhibitors. The remaining authors declare no competing interests.

## Additional information

[1]Lupus Clinical Trials Unit, National Institute of Arthritis and Musculoskeletal and Skin Diseases (NIAMS), National Institutes of Health (NIH), Bethesda, MD, USA. [2]Systemic Autoimmunity Branch, NIAMS, NIH, Bethesda, MD, USA. [3]Section of Inflammation and Cardiometabolic Diseases, National Heart Lung and Blood Institute (NHLBI), NIH, Bethesda, MD, USA. [4]NIH Clinical Center Biostatistics and Clinical Epidemiology Service, Bethesda, MD, USA. [5]Office of the Clinical Director, NIAMS, NIH, Bethesda, MD, USA. [6]Translational Immunology Section, NIAMS, NIH, Bethesda, MD, USA. [7]Arthritis and Pain Associates of PG County, Greenbelt, MD, USA. [8]Trans-NIH Center for Human Immunology, Autoimmunity and Inflammation, NIH, Bethesda, MD, USA. [9]Hematology Branch, NHLBI, NIH, Bethesda, MD, USA. [10]Biodata Mining and Discovery Section, NIAMS, NIH, Bethesda, MD, USA. [11]Feinstein Institute for Medical Research, Manhasset, NY, USA. [12]Translational Vascular Medicine Branch, NHLBI, NIH, Bethesda, MD, USA. [13]Molecular Immunology and Inflammation Branch, NIAMS, NIH, Bethesda, MD, USA. ✉email: hasnisa@mail.nih.gov

