## [Peer Review File · Nature Communications]

Response to Reviewers:

On behalf of all authors of the manuscript thank you for giving us the opportunity to resubmit the revised manuscript. Please see below our point by point response and the manuscript have been with all changes highlighted in yellow.

Reviewer #1:

Remarks to the Author:

In the present study, Hasni et al evaluated the effects of tofacitinib on cardiometabolic and immunologic markers associated with atherosclerosis in SLE patients. This study was double blind, placebo controlled and patients were stratified by the presence of a STAT4 risk allele. The primary outcome was to determine safety and tolerability of tofacitinib in subjects with mild-moderate SLE disease activity. The secondary outcomes, according to the authors, included clinical efficacy or no worsening of disease activity, effects on quality of life measures and others. In addition, vascular variables, NETs and gene expression studies were done and plasma lipoproteins measured. Based on the results, the authors concluded that tofacitinib improved immunologic and cardiometabolic markers.

The study is very small and the differences shown only spurious and possibly a play of chance, given the lack of a predetermined, hypothesis driven hierarchical statistical approach and no statistical correction regarding multiple assessments. Moreover, and more importantly, no clinical effects were observed.

1. At the clinicaltrials.gov site of the study, the authors state that “Identifying a drug that has immunomodulatory effects and is also vasculoprotective is an unmet need in this disease.” The fact, that disease activity did not improve and apparently (not reported) autoantibody and complement levels did not change, makes tofacitinib inappropriate for use in SLE, irrespective whether or not some minor changes in certain cardiovascular variables are found.
2. Further, one of the secondary endpoint according to clinical trials.gov was “Assessments of clinical response” and not, as the authors write on p. 2: “clinical efficacy or no worsening of the disease activity“. As stated above, there was no clinical improvement seen in either group.
3. It is noteworthy, that all clinical characteristics (except for skin abnormalities) showed higher activity in the placebo compared with the active group and fatigue was highly significantly more prominent in the placebo population (suppl. Table 3), fatigue being a clinical though subjective marker of SLE activity. In line, lipid levels were lower (given the small numbers essentially significantly lower) in the placebo population (Suppl. Table 2). Thus, the populations were not balanced despite randomization, a consequence of the small patient number. Data on SLE pts with and without the risk allele are not shown, so further assessment of such potential differences regarding some of the study results cannot be done. Overall, the active treatment group may have had to low a disease activity and, therefore, may not have been the appropriate population for this study (in other clinical trials the SLEDAI-2K at baseline is usually twice as high).
4. A decrease in pSTAT1 expression in CD4+ cells is reported on tofacitinib but not placebo. However, when looking at Fig. 1 a and b, the data at baseline and day 56 are almost identical among the two arms and the lack of significance in the placebo group is likely due to the lower number of patients (10 vs 20), in line with the weak statistical significance of $p=0.02$ for the tofa group). Given

that pSTAT1 expression was not affected in any other cell population tested nor other JAKs affected even in CD4+ cells, this is likely a play of chance (see above) and, as also mentioned, the curves are quite similar for the two arms.

5. Changes in lipid levels and variables related to atherosclerosis have been previously reported for JAK inhibitors in other diseases – making similar observations here is not surprising. Alas, the consequences of this finding would have to be assessed in long-term outcomes studies in SLE patients which are unlikely to be undertaken given the lack of clinical response.

Minor points

1. The authors also show downregulation of IFN stimulated genes by tofacitinib. This is an expected observation, but since it is not associated with clinical improvement, it must be regarded as pharmacologic epiphenomenon. Given these findings, the authors could have discussed the enigmatic role of the IFN signature in SLE.

2. The data related to the risk allele are interesting, alas, in light of lacking clinical associations they are not very meaningful

3. A figure legend seems to be missing.

Response to reviewer #1:

Thank you for your comments. Please see below our point by point response to your comments:

- This was a phase Ib study to explore if tofacitinib is safe and not associated with any serious side effects. We concur that it cannot be concluded that this drug will be efficacious in SLE since the study was not powered to assess clinical efficacy but, rather, to assess safety, tolerability and potential modulation of immune and cardiometabolic parameters. Therefore, larger and long-term studies of tofacitinib use in SLE are needed to determine its putative clinical use. This is supported by the encouraging results from the phase II study using another Jak inhibitor, baricitinib, in SLE, pointing toward a putative role of JAK-STAT pathway inhibition in SLE.
- One of the secondary aims was to explore if tofacitinib use was associated with clinical efficacy. The main purpose of this was to make sure that, by blocking for example IL-2, we would not aggravate SLE disease activity. The study was not powered for efficacy and, therefore, given the relatively low SLEDAI disease activity at baseline, we were not expecting to see significant improvements in clinical disease. We have revised the abstract and the main manuscript as below to accurately reflect our secondary endpoint:

Abstract

The secondary outcomes included assessment of clinical response, as measured by the Systemic Lupus Erythematosus Disease Activity Index 2000 (SLEDAI-2K), physician global index (PGA), the British Isles Lupus Assessment Group disease activity index (BILAG), and effects on quality of life measures (Short Form Health Survey (SF 36) and Multidimensional Assessment of Fatigue questionnaire (MD-Fatigue).

Main manuscript

The secondary outcomes included assessment of clinical response, effects on quality of life measures and several exploratory mechanistic studies to evaluate effect of the drug on

immune dysregulation and cardiometabolic signatures associated with the development of premature cardiovascular disease.

- The clinical trial was a phase I study designed primarily to assess safety of tofacitinib in SLE. While there may have been some variation in matched baseline characteristics between placebo and drug groups, SLEDAI-2K, DAS28, CLASI, and PGA were not statistically significantly different between the two arms (Suppl. Table 3). The fatigue was indeed higher in subjects on placebo at the baseline, and we have updated the manuscript to reflect this difference by adding the following statement:
Despite randomization, there were some differences between tofacitinib and placebo group at the baseline visit. The subjects on placebo reported a statistically significant higher degree of fatigue, as measured by MD-Fatigue scale at baseline and a significant improvement in fatigue during the study. The subjects on placebo had higher baseline DAS-28-ESR and lower baseline anti-ds-DNA levels as compared to tofacitinib group, but these were not statistically significant differences (Table 4).
- The total cholesterol was lower in the placebo group at the baseline but this was not statistically significant and it appeared to be mostly related to a lower triglyceride in placebo group. The lipid values remained mostly unchanged throughout the study, except for the changes in HDL at Day 56. The outcomes of significance i.e. HDL and LDL were not significantly different in the placebo vs tofacitinib groups at the baseline. We have added the following statement to the manuscript to reflect this:
Compared to tofacitinib group, the placebo group had lower cholesterol and triglyceride values at the baseline, but these differences were not statistically significant and remained essentially unchanged throughout the study (Table 3).
- The data on SLE patients with or without STAT 4 risk allele is now added as Supplemental tables 1 and 2, along with the following statement added to the results section:
The safety and disease activity data of subjects on tofacitinib and placebo stratified based on presence or absence of *STAT4* risk allele revealed non-statistically significant differences in some variables between groups (except for as noted elsewhere); the clinical relevance of these differences are uncertain and needs to be further explored in a larger sample (supplemental tables 1 and 2).
- We agree with the reviewer's comment about the active group having low disease activity. This was as per the inclusion criteria of the trial, which was restricted to subjects with mild-severe but stable disease activity on HCQ and prednisone only. The primary objective of the trial was to assess safety of tofacitinib in SLE and larger population with higher baseline disease activity are needed for an efficacy study.
- The fold changes in pSTAT1 in stimulated/unstimulated CD4⁺ cells in subjects on tofacitinib was significantly lower at day 14 and day 56 with return to baseline at day 84. These changes reflect the expected, known biological effect of tofacitinib based on studies from our group and others. There is indeed a downward but not significant trend in placebo group with no plausible explanation. We agree with the reviewer that the changes could be more robust with a larger sample size. We plan to further explore the pharmacodynamic effects of tofacitinib in much larger populations. We have added the following to results:

There was also a trend observed towards nonsignificant inhibition of pSTAT1 in subjects on placebo that is of unknown significance and without any known plausible biological explanation.

- Changes in the lipid levels have been associated with JAK inhibitors as a class effect. What is significantly different and an additional advance in this study is the effect of tofacitinib on HDL numbers, particle size, HDL function (cholesterol efflux) and the modulation of LCAT activity. We are planning to explore further these effects of tofacitinib on lipoprotein subfractions in a long-term study in patients with SLE.
- We have modified our discussion to underscore the point raised by the reviewer. We do not consider this a pharmacologic epiphenomenon because, while the subjects were clinically fairly quiescent, it is very likely that smoldering disease and immune dysregulation can contribute to chronic complications. The paragraph now reads:
While tofacitinib affected type I IFN responses and markers of neutrophil dysregulation, the effects observed on other immune cell types or **clinical activity** in subjects with mild to moderate SLE were not significant. Future studies on the effects of tofacitinib in larger patient groups, stratified by genetic risk and greater active disease, will be needed to assess alteration of both innate and adaptive immune parameters, and **clinical disease activity**.
- This is a first study to our knowledge where patients were stratified based on a genetic risk allele. We agree that further studies with larger patient population will be needed to ascertain its significance. However, the associations with distinct cardiometabolic parameters is of significant interest and, based on several studies we cited in our manuscript, represent an important attempt to address how specific gene polymorphisms modulate disease. For example, the association of STAT4 risk allele with enhanced NET formation adds to recent data that genetic determinants can promote enhanced generation of autoantigen and contribute to tissue damage.
- Figure legend is added at the end of the main manuscript and as a separate file for the reviewers.

Reviewer #2:

Remarks to the Author:

This is a report of an early phase trial of the drug tofacitinib in SLE.

The primary outcome of this small trial was to determine safety and tolerability in subjects with mild-moderate SLE activity.

The authors have provided a copy of the full protocol and statistical analysis plan with the submission.

The results of this safety and tolerability trial have been over interpreted with the exploratory analysis of the secondary outcomes given high prominence in the abstract and title. Though CONSORT has been followed for the primary outcome no summary statistics and 95% confidence intervals have accompanied any of the conclusive statements in the abstract.

There are some nice figures for some of the results but the figures do not have accompanying legends

and it is not clear what the error bars represent.

Revisions necessary:

1. The conclusions in the abstract need to be toned down to reflect that many secondary outcomes were analysed and no correction of multiple testing was performed. More quantitative evidence needs to be reported in the abstract. Hence a statement such as 'there was no worsening of serological SLE disease activity...' needs to be accompanied with a summary statistic and a 95%CI.
2. There is an over reliance on the use of p-values to decide if a result is 'significant' or not, almost no p-values above 0.05 are reported. If p-values are to be used in the exploratory analysis then all calculated p-values should be clearly reported.
3. The description of the statistical analysis used is inadequate. Readers should not have to look at the supplementary material to find out what methods has been used to analyse the data in the statistical analysis plan. Even when examining the statistical analysis plan It is not fully clear what statistical methods have been used where and when.
4. The CONSORT diagram should be in the main paper not the supplementary materials.
5. I cannot see any legends for figures 1-4. How have the comparisons between distinct times been performed? what do the error bars represent in the figures?
6. More effort should be made to show key aspects of the data in the main paper that relates to the primary outcome. Saying that no statistically significant differences were seen (lines 176 to 190) is not very reassuring when this study has not been powered to detect differences.
7. On lines 254 to 264 it would be easier to view means and 95%CIs than means +/- SEs. It would also be preferable to present full results in a table, which can be read alongside the graph rather than listing them within a paragraph.

Response to reviewer #2:

Thank you for your comments. Please see below our point by point response to your comments:

- We have revised the abstract and methods section to mention exploratory nature of some of the analyses:

In addition, we performed assessments of large, medium and small vascular function, measured plasma lipoproteins, neutrophil extracellular traps (NETs), immune cell subsets and their gene expression, to explore if Jakinibs improve cardiometabolic and immunologic parameters associated with enhanced CVD risk in SLE.

Some of these improvements were more robust in SLE subjects with *STAT4*-risk allele.

- We have revised the abstract with the following language to provide summary statistics and 95% CI and mention that no corrections were made for multiple testing:

There was no worsening of clinical activity, with a difference in change scores SLEDAI 2K (tofacitinib vs. placebo) of 0.04 (95 % CI: -1.04,1.11) at day 56 and -0.72(95% CI: -1.98,0.53) at day 84. None of these differences were statistically significant. There were no new BILAG A or B scores or worsening of serological SLE disease activity and patient reported outcomes. Compared to placebo, tofacitinib treatment led to improvements at day 56 in high density lipoprotein cholesterol levels(p=0.0006, CI 95%: 4.12,13.32) and particle number(p=0.0008, CI 95%: 1.58,5.33); lecithin: cholesterol acyltransferase concentration (p=0.024, CI 95%: 1.1, - 26.5)

and cholesterol efflux capacity($p=0.08$, CI 95%: -0.01, 0.24). There were also improvements in arterial stiffness and endothelium-dependent vasorelaxation in the tofacitinib group.

- We have revised the results section to mention p-values above 0.05, in addition p-values with 95% CI were added in the manuscript and the tables.
- We have revised the statistical methods section as below:

Statistical Analysis:

The sample size chosen was based on what is commonly being used in similar studies and consistent with our experiences in early phase safety studies(49, 50). No formal power calculations were performed. Data from all randomized subjects were included in the analysis. The adverse events were summarized with frequency counts and percentages for each treatment group. To evaluate the treatment effect on the change from baseline to the end of treatment period (Day 56), a linear mixed effects model approach was utilized to fit the longitudinal data on continuous safety parameters and clinical outcomes. The model included the baseline value, the *STAT4* risk allele status, the treatment group, the categorical time point, and the treatment by time interaction as fixed effects. An unstructured variance covariance matrix was used to account for the correlations among repeated measures. For variables assessed only at baseline and Day 56 during the treatment period, the analysis of covariance (ANCOVA) models were used including baseline, treatment, and the *STAT4* risk allele status as the covariates. In addition, paired t-test, Mann-Whitney u or ANOVA were used for comparison where appropriate based on normality of distribution. Change scores from baseline to Day 84 were analyzed separately using ANCOVA models. No adjustments for multiple comparisons were made to the p-values due to the exploratory nature of analysis. First patient was enrolled on July 28th, 2016 and the last patient was enrolled on September 18th, 2017. All statistical analyses were performed with SAS software (version 9.4).

RNA sequencing analysis: Gene expression values were calculated with Partek Genomics Suite 6.6, which was also used for the principal components analysis (PCA) and 1-way ANOVA. The ANOVA was performed on log₂ transformed RPKM with a 0,1 offset.

NanoString data analysis: Assessment of quality of the runs was done (supplementary methods). Data were combined, normalized, and analyzed in Excel (Microsoft Corporation; Redmond, WA). JMP version 14 was used for further statistical analysis and plotting (SAS Corporation; Cary, NC). Synthetic DNA oligonucleotides of each of the 37 ISGs and 4 housekeeping genes were used as a calibration standard to check run and reagent lot consistency.

- We have added the CONSORT diagram to the main paper
- Figure legend is added at the end of the main manuscript and as a separate file for the reviewers.
- We have modified the results section to show key aspects of the data:

There were significantly more African American subjects in the *STAT4* risk allele negative subgroup in both tofacitinib and placebo groups($p=0.002$). The subjects with *STAT4* risk allele were younger and mostly Hispanic, but these differences were not statistically significant. There were 71 adverse events: 43 in the tofacitinib group and 28 in the placebo group, with no serious AEs in the tofacitinib group (Table 2). Most of the AEs observed in the tofacitinib group were mild (16/43) and moderate (5/43) upper respiratory infections that either self-resolved or after treatment with oral antibiotics. No herpes zoster reactivation, BK viremia, or venous thromboembolic events were recorded. There were no clinical or statistically significant changes

observed in the tofacitinib group compared to baseline measurements and to the placebo group in other laboratory safety parameters (Table 3). As compared to placebo in the tofacitinib group, the hemoglobin difference in change score was -0.33 (95% CI -0.33,-0.88) at day 56, -0.20 (95% CI, -0.81,0.40) at day 84; white blood cell count difference in change score was -0.63(95% CI -0.63,-1.46) at day 56, -0.52 (95% CI, -1.35,0.30) at day 84; absolute neutrophil count difference in change score was -0.58(95% CI -0.58,-1.35) at day 56, -0.21 (95% CI, -0.85,0.43) at day 84; platelet count difference in change score was -15.36(95% CI -15.36,-35.79) at day 56, 5.3 (95% CI, -12.33,22.93) at day 84; serum AST difference in change score was 0.71(95% CI, 0.71,-5.14) at day 56, -13.8 (95% CI, -34.91,7.31) at day 84; and serum ALT difference in change score was 1.58(95% CI, 1.58,-4.55) at day 56, -2.15 (95% CI, -8.86,4.56) at day 84 (Table 3). None of these differences were statistically significant. None of the patients in either group met the disease flare criteria during the trial and there were no new BILAG2004 A or B scores. The baseline mean SLEDAI 2K score in the tofacitinib group was 5.1 ± 2.2 mean \pm standard deviation, compared to 5.5 ± 3.7 in the placebo group; the difference in change scores (tofacitinib vs. placebo) was 0.04 (95% CI: -1.04,1.11) at day 56 and -0.72(95% CI: -1.98,0.53) at day 84. The baseline mean BILAG 2004 score in the tofacitinib group was 7.6 ± 4.6 mean \pm standard deviation, compared to 9.3 ± 4.3 in the placebo group; the difference in change scores (tofacitinib vs. placebo) was 1.56 (95% CI: -1.86, 4.98) at day 56 and -2.04 (95% CI: -4.96,0.89) at day 84. The baseline mean PGA score in the tofacitinib group was 2.5 ± 2.8 mean \pm standard deviation, compared to 3.9 ± 2.8 in placebo group; the difference in change scores (tofacitinib vs. placebo) was 0.60 (95% CI: -0.93, 2.13) at day 56 and 0.76 (95% CI: -1.01, 2.53) at day 84. None of these differences were statistically significant. The SLE serological disease activity (complement C3 and C4 levels) and the patient reported outcomes (SF-36) were similar at baseline visit and did not have a significant difference in change scores at day 56 or day 84 (Table 4). This study was not powered to assess clinical efficacy.

- Full results are added in the respective tables. We have revised lines 254-264 as below:
At day 56, a significant increase in Lecithin-cholesterol acyltransferase (LCAT) concentration was detected in SLE subjects on tofacitinib that were *STAT4* risk allele-positive ($p=0.024$, 95% CI: 1.1 - 26.5) compared to the *STAT4* risk allele-positive group on placebo or in subjects negative for this risk allele on tofacitinib ($p=0.04$, 95% CI: -18.2,10.2) (Figure 3b). In addition, at day 56, there was a statistically significant increase in cholesterol efflux capacity in SLE subjects treated with tofacitinib ($p=0.002$, 95% CI: 0.04 to 0.16) and a non-significant trend when compared to placebo ($p=0.08$, 95% CI: -0.01, 0.24) (Supplemental Figure 5e). This effect was not dependent on the subject's *STAT4* risk allele status. No statistically significant differences in insulin resistance, as measured by HOMA-IR, were detected between placebo and tofacitinib-treated groups ($p=0.51$, 95% CI: -0.93, 0.48) (Table 3). Compared to tofacitinib group, the placebo group had lower cholesterol and triglyceride values at the baseline, but these differences were not statistically significant and remained essentially unchanged throughout the study (Table 3).

Reviewer #3:

Remarks to the Author:

In the present paper, the authors describe the results of a randomized, double-blind, placebo-controlled clinical trial of tofacitinib (TFC) in patients with systemic lupus erythematosus (SLE). The duration of the trial was 56 days followed by 28 days off-study drug period. The main aim of the trial is to determine safety and tolerability of TFC. As secondary outcomes, evaluate the efficacy through several measures, including the assessment of vascular and immunological function. It represent the first trial to date that

explores the role of tofacitinib in SLE, providing also interesting information about the effect of TFC on vascular function, in a moment when there is increasing concern about the safety of the drug since the FDA warning regarding thromboembolic risk and mortality.

TITLE:

The title of the study is correct, clear and informative.

ABSTRACT:

The abstract is also correct. The authors provide enough background information and gap in the knowledge, so as the reader can identify the motivation of the authors to carry on this study. Also, they report an adequate summary of the methods employed that give enough information to the reader. The key results of the study are included in the abstract and so are the conclusions of the authors.

INTRODUCTION:

The authors provide adequate background information in order to put the reader in context, supported by compelling bibliography. They also provide the reader with evidence available before the study, illustrate the reader with their opinion on what do this study add to the current knowledge and what implications it may have. Globally, it is well written and informative of the purpose of the study. However, I have some comments that I would appreciate if the authors could respond:

- In line 99, page 4, the authors provide a reference a little old-fashioned, yet valid, regarding the epidemiology of the cardiovascular risk in SLE. Maybe the authors can consider to add/substitute with other reference more recent, published in a journal as good as the chosen: Giannelou M, Mavragani CP. Cardiovascular disease in systemic lupus erythematosus: A comprehensive update. *J Autoimmun.* 2017;82:1–12. doi:10.1016/j.jaut.2017.05.008.
- In line 110, page 4, when the authors state “Many inflammatory cytokines implicated in SLE pathogenesis, including type I and II interferons (IFNs), signal through a JAK-STAT pathway”, the reference they provide [8] (Dean GS, Tyrrell-Price J, Crawley E, Isenberg DA. Cytokines and systemic lupus erythematosus. *Ann Rheum Dis.* 2000;59(4):243-51) does not mention anything about the JAK/STAT pathway nor how do the INFs transduce signals. Maybe the authors have mistaken the reference. As a suggestion: Alunno A, Padjen I, Fanouriakis A, Boumpas DT. Pathogenic and Therapeutic Relevance of JAK/STAT Signaling in Systemic Lupus Erythematosus: Integration of Distinct Inflammatory Pathways and the Prospect of Their Inhibition with an Oral Agent. *Cells.* 2019;8(8):898. Published 2019 Aug 15. doi:10.3390/cells8080898. However, they are lots of papers regarding this issue, and surely the authors are familiar with them and could choose another if they want.
- In line 118, page 5, the authors describe their study as a phase Ib/IIa. To my knowledge, phase I trials aim is to define the treatment’s safety, determine a safe dosage range, and explore the drug’s metabolism, and pharmacokinetic and pharmacodynamic profile, usually conducted in healthy volunteers. On the other hand, phase II studies are usually conducted in a larger group of subjects to determine the treatment’s efficacy and further evaluate safety. When divided into IIa and IIb, phase IIa trials are specifically designed to assess dosing. The term “phase I/II” is used when a trial combines both phases, in order to perform a more rapid investigation. In the current paper, the authors conduct a trial where no different doses are studied, and the sample of patients is quite reduced. In a previous study of another JAK-inhibitor (Baricitinib), phase II, 314 patients were enrolled, and different doses were

studied. In addition, in www.clinicaltrials.gov, NCT02535689 is registered as a phase Ib study. My doubt is whether the term “phase II” would be correct in this case, given the low number of patients recruited and the absence of different doses. Maybe the authors could clarify this aspect.

- In line 122, page 5, wouldn't it be more accurate to state that “STAT4 risk allele has been suggested to increase the PRODUCTION of type I IFN” instead of RESPONSE to type I IFN?

METHODS:

- In lines 131-132, page 5, SLEDAI scores greater than 10 imply high disease activity, so it would be more correct to state mild to severe disease activity, instead of mild to moderate (0-10 in SLEDAI score).

- In lines 135-136, authors describe that patients were on “stable doses”. How many time did the authors consider as stable? Is the same for antimalarials than for glucocorticoids?

- What is the reason behind allowing such higher doses of glucocorticoids (20 mg/d of prednisone or equivalent) but no immunosuppressants, specially taking into account that patients may have severe SLE as determine by SLEDAI scores (up to 14)? This is important, since it is not adjusted to clinical routine. Patients with high disease activity (SLEDAI scores greater than 10) probably require more therapy than antimalarials and high doses of prednisone, since as the authors are aware, such high doses of glucocorticoids are associated to significant morbidity and even mortality (in fact of a cardiovascular origin, among others). It is true that the mean SLEDAI is low-moderate (5.2 and 4.9 in the tofacitinib group, and 5.6 and 5.2 in the placebo group) and the mean PDN dose is also moderate (5mg/d - 7.5 mg/d), but, as described earlier in the methods section, some patients may be included in the trial with high disease activity and an unusual therapy (only antimalarials and very high doses of prednisone, as just explained before). Do the authors consider that this fact may introduce a selection bias, or may limit the patients to whom extrapolate the results? How many patients were included with such characteristics?

RESULTS:

- In line 189-190, page 8, the phrase “overall, these results indicate that tofacitinib was well tolerated in SLE” is an interpretation made by the authors of the results obtained, and therefore maybe could be better placed in the discussion section instead.

- In figure 1c, candidate genes in the Tofacitinib group at day 84 seem to be still downregulated with respect to placebo group. Are this differences observed meaningful?

- In lines 210-212, page 9, the phrase “overall, these results indicate that tofacitinib modulated the type I IFN pathway and effectively decreased pSTAT1 levels in CD4+ T cells” is an interpretation made by the authors of the results obtained, and therefore maybe could be better placed in the discussion section instead.

- According to the results obtained by the authors, low density granulocytes (LDGs) were significantly reduced in the Tofacitinib group at both 56 and 84 days. The mean life span of a granulocyte oscillates between few hours and few days. However, also according to authors, gene expression at day 84 had returned to baseline in the Tofacitinib group. What is the explanation of the authors of this reduction in percentage of the LDG?

DISCUSSION:

The discussion is well written and the authors have made a great job linking their results with the

existing knowledge.

Response to reviewer #3:

Thank you for your comments. Please see below our point by point response to your comments:

- We have added the references mentioned by the reviewers.
- We agree with the reviewer's comment about different phases of clinical trials. However, since the study design does not fit the definition of one phase i.e. some parts of the study are more of phase I(to test safety and side-effects) and some are more phase IIa(pilot studies designed to demonstrate clinical efficacy or biological activity, 'proof of concept' studies) we choose to use the term phase I/II.
- There is evidence to suggest that *STAT4* risk allele increase production and sensitivity to type I IFN in SLE. As shown by Kariuki et al; Journal of Immunology 2009, Autoimmune disease risk variants in *STAT4* can confer increased sensitivity to IFN-alpha in lupus patients. We have also added this reference. The introduction section has been modified as below:

As presence of the *STAT4* risk allele (rs7574865) has been associated with more severe clinical phenotype and significant increased risk of vascular disease in SLE, and since type I IFNs activate *STAT4*, we stratified subjects based on the presence(+) or absence (-) of *STAT4* risk allele which has been suggested to increase the production of and sensitivity to type I IFNs in peripheral blood mononuclear cells of SLE patients, to investigate the effect(s) of these haplotypes on the clinical and immunologic response to tofacitinib.

- We agree with the reviewer about the SLEDAI cut-off and changed the wording as follows:
Thirty adult SLE subjects that met American College of Rheumatology (ACR) Revised Criteria for the Classification of SLE and had mild to severe disease activity (Systemic Lupus Erythematosus Disease Activity Index 2000 (SLEDAI 2K) score between 2-14) were enrolled in an outpatient clinical research setting.
- For glucocorticoid the dose must be stable for the 4 weeks prior to screening visit. For anti-malarials the dose must had been stable for the 12 weeks prior to the screening visit. We have revised the methods section as follows:
Eligible subjects were on stable doses of anti-malarials (for 12 weeks prior to the screening visit) and/or oral glucocorticoids (for 4 weeks prior to the screening visit) (prednisone or equivalent < 20 mg/day) but no immunosuppressants were allowed.
- We allowed patients who may be on high dose of steroids due to a flare and put on tofacitinib to assess its clinical efficacy and possible steroid sparing effect. In addition, we wanted to avoid any additional immunosuppressive agents due to potential of side effects such as infections when tofacitinib is added to the regimen. We agree that the results from this study may not be extrapolated to such patients. However, our intent with this study was to show the safety of tofacitinib in SLE, further studies are needed in larger patient population with more severe disease and in combination with other immunosuppressive to establish its efficacy (as mentioned in discussion section).
- We agree with the reviewer and have removed the statements mentioned form the results section, and mentioned them under discussion.

- Even though most of the candidate genes returned to baseline, a few genes were still significantly downregulated. We have updated the results section to reflect this as below:

By day 84, the levels of most of these 19 ISGs had returned to pretreatment levels, but a few genes were still significantly downregulated (Figure 1c and Supplemental Figure 3).

Nanostring was used to verify the impact of tofacitinib on the IFN signature (Figure 1c). The IFN response gene score was not significantly different at baseline between subjects on tofacitinib and placebo. At day 56, the subjects on tofacitinib had a significant reduction in levels of ISGs in comparison to placebo ($p=0.01$), which remained significant at the day 84 visit ($p=0.02$) (Supplemental Figure 4)

- At this point we do not have an exact mechanism to explain why tofacitinib reduced selectively the LDG counts and its effect extending to day 84. One possibility is that the inhibition of the effect of certain cytokines on LDGs either prevented their release from bone marrow and/or facilitated apoptosis in tissues or in the bone marrow. Future studies will investigate this further. We have added the following in the discussion:

The mechanism by which tofacitinib led to decreases in LDGs remains to be further characterized and may be related to inhibition of cytokine-specific effects on these cells in the bone marrow or tissues, perhaps promoting their death through apoptosis.

Reviewer #4:

Remarks to the Author:

Key Results: This paper reports the results of an early phase study of the Janus kinase inhibitor, tofacitinib for SLE. This was a double blind, randomized trial in which 20 subjects received tofacitinib and 10 received placebo. In each group, half of the subjects were homozygous for the STAT4 allele that has been associated with a severe clinical phenotype of SLE with increased risk for vascular disease. This paper briefly summarizes a distilled version of primary and secondary endpoints (see issues with inaccurate representation of protocol endpoints below) and focuses almost entirely on a very interesting and potentially important exploratory analysis, integrating some components which are described as secondary endpoints in some parts of the protocol but with biomarker components described as exploratory in the statistical plan. The presence of the STAT4 risk allele is evaluated against immunologic and cardiac risk features in those treated with tofacitinib vs placebo. Since STAT4 is activated by type I interferons much of the focus in this report is on immunologic variables associated with type I interferon pathways. Tofacitinib use was associated with decreased type I interferon response gene score, and patients with the STAT 4 risk allele had significantly higher levels of circulating neutrophil NET complexes at baseline, which decreased significantly with tofacitinib, not found in subjects receiving placebo or those without the risk allele. It is unclear in what hierarchical order these analyses were performed, since it appears that a wider unsupervised data analysis included the evaluation of an unspecified range of cytokines and cell types which showed no differences in treatment groups. Additionally, the following reversible cardiovascular risk markers were improved in the tofacitinib-treated group: HDLc, LCAT (this one only in the STAT4 risk allele group), cholesterol efflux capacity, arterial stiffness, and endothelium-dependent vasorelaxation.

This paper reports that the primary endpoint of safety was met as were secondary endpoints of "no worsening." No adjustments for multiple comparisons were made and the authors acknowledge that this study was not powered to assess clinical efficacy. It is unclear what was "met." Only one supplementary table is provided to summarize the topline results, however this report is vague in how it defines these endpoints. In fact, the primary endpoint as defined in the protocol and statistical plan included no differences in adverse events and in flares. There are two issues with this flare component: first it is not even acknowledged in this trial report. Second, the flare definition provided in the protocol is a greatly truncated version of an accepted (albeit problematic) flare instrument which is likely to miss most clinically significant flares of disease. It is problematic that this paper reports that the primary endpoint was "met" when there is no data at all provided about flares. Similarly secondary endpoints are reported as met based on the fact that this small population with evidently mild disease (mean SLEDAIs < 6) did not have any change in disease activity throughout a short trial. This appears to be an exaggeration of the actual scope of data. This paper has an exciting exploratory premise and findings. It probably should not try to also serve as an inaccurate topline report of a clinical trial or as justification for the overly conclusive language used in the title, abstract and discussion about the type I interferon, Stat4, or cardiovascular risk findings.

Data Approach and Methodology: The technical approach to biomarkers seems to be a strength of the paper, for example NanoString was used to verify the impact of tofacitinib on the interferon signature. Indeed, if the scope being reported was simply this exploratory biomarker and clinical cardiac risk analysis, with more immediate clarity that this was indeed an exploratory analysis, it would strengthen the paper greatly. The main concern about the methodology is the lack of clarity in the approach to analysis. The title and the way the abstract is written implies that the main purpose of this project was to determine whether JAK inhibition in SLE subjects might improve cardiometabolic and immunologic parameters associated with enhanced CVD risk. Although it was true that this hypothesis can be inferred from the original protocol, it was nowhere near the primary or major secondary objectives or endpoints for the trial. The abstract is therefore misleading and should read something more like "We utilized data and a biomarker exploration in a small Phase 1 trial of tofacitinib to test the hypothesis that that JAK inhibition in SLE subjects might improve cardiometabolic and immunologic parameters associated with enhanced CVD risk." This would clarify from the beginning that multiple comparisons were made and the analysis is exploratory without detracting from its high interest. The paper may be suffering from an identity crisis. Is this the first report of a Phase 1b study? If so, why are the primary and secondary endpoint results relegated to supplementary tables? And why are the primary and secondary endpoints not defined in the paper the way they are in the protocol and statistical plan (which also contain internal discrepancies, particularly about whether biomarker studies are secondary endpoints or exploratory endpoints. This is substandard reporting and detracts from the very compelling exploratory biomarker and cardiovascular risk analysis.

Appropriate Use of Statistics and Treatment of Uncertainties: The statistical analysis plan seems ambitious for the size and length of the study. This study only includes 20 treated patients and 10 placebo patients. It is explained that the study was modeled after a Phase I safety study. However, Phase I safety studies are not powered for subset analysis of patients stratified by a genetic marker. Therefore, wherever there is a lack of finding, and wherever there are findings with borderline statistical confirmation this is not really interpretable. Although the paper does, in some parts of the discussion

acknowledge the preliminary and exploratory nature of the data, most properly reported pilot studies are less decisive in conclusions. For example saying something like "no safety signal was seen in this preliminary study" would be more accurate than claiming that the trial met its primary and secondary endpoints of safety and no worsening of disease, especially when they are not reported according to the provided statistical plan and which were defined with an extremely low bar for claiming success.

Conclusions: The findings are interesting but they are at best hypothesis-generating and should not be overinterpreted. For example, a statement that "Results from the current trial showed the effect of JAK inhibition was more robust in subjects with STAT4 risk allele," is unsupported. The findings in this report are interesting, potentially important and worth further study, but, because they are exploratory and subject to uncertainties from multiple analyses, they are not robust. Furthermore nothing has been "shown." "Suggested" might be a better term than "shown." Furthermore, even reaching the conclusion that the effect of JAK inhibition was greater in the STAT4 risk allele group, when only certain features were found to be more responsive, is an overstatement. Finally the statement that no other drug has been reported to have beneficial cardiovascular impact in SLE is incorrect. Too many to list here (see PubMed).

Population imbalances should also be discussed. From the data provided, the STAT 4 risk allele positive patients in this small study were younger with less disease duration and most were Hispanic. Most of the STAT 4 risk allele negative patients were of African descent. It needs to be acknowledged that many pathways converge on STAT 4 and in a study this size it may not be clear what is cause or effect vs chicken and egg.

Other comments on Interpretation of results:

1. In Fig 1a AND 1b the STAT 1 result does not look very different, so some discussion of unknown clinical significance of differences should be entertained.
2. Fig 2b also shows a trend to decreased total neutrophils in those randomized to tofacitinib. This might be mentioned in the discussion.
3. In Table 1 there is a greater percentage of mild and more moderate infections in the treatment group as compared to placebo. In a small study underpowered to show statistical differences in events that may indeed be more common this should be reported as something other than "the primary endpoint of safety was met."
4. Supplemental Table 3 (if even appropriate to report primary and secondary outcomes as a supplementary finding) should be organized by the original definitions for primary and secondary endpoints that were pre-specified in the protocol, since very little about these is described in the main part of the paper. The primary endpoint should include the prespecified flare analysis regardless of whether it seems unlikely any differences were found. However, the potential for misleading data due to the truncated flare definition used should also be acknowledged. A better way to capture flares (perhaps as an additional exploratory analysis) would be to use the BILAG data that was fully and sufficiently collected to assess flares. Current guidelines for BILAG flare defines moderate flare as more than 1 B score based on at least one B-fulfilling parameter rated "worse" or "new." Definition of severe flare is any A score based on 1 or more A-fulfilling variables rated "worse" or "new."

(validated in Isenberg Ann Rheum Dis. 2011 70:54-9).

5. It should be acknowledged, at least in the figure legend to Supplemental Table 3, that the Multidimensional Assessment of Fatigue questionnaire showed a small trend to worsening in the Tofacitinib-treated group and improved in placebo group. Those randomized to tofacitinib were also less fatigued at baseline

6. The results of the Physician Global Assessment in Supplemental Table 3 are difficult to interpret. Please give the units of PGA. It seems unlikely by the data that this was a 100 mm scale and very likely that it was the 1-3 scale used in most lupus trials. Therefore, how could placebo group have a mean PGA of 3.9 at Day 1?

7. DAS 28 = Disease Activity Score of the 28 joints may be a little worse in the placebo group as per Table 3. You need to define the DAS 28 used. Were CRP or ESR included?

Originality and significance: The main significance is that a model for further testing of tofacitinib effects on cardiovascular variables looks promising and should be tested.

Suggested improvements: This paper would be strengthened if it were organized in a more straightforward writing style, clarifying from the beginning that this is a small preliminary examination of tofacitinib in SLE with an exploratory analysis of cardiovascular markers. Or report the clinical findings more properly in a separate brief report and make this paper an exploratory analysis of cardiovascular markers from a pilot examination of tofacitinib in SLE. The findings remain very interesting, and should be studied further. Validation in a new patient population to test what would now be the major, prespecified hypotheses, would greatly strengthen this paper. Barring that, a more modest report of these findings as preliminary, exploratory cardiovascular and type I interferon results from an inconclusive pilot trial could be warranted.

Clarity and context: The report is well written. The only comment on clarity is, again, the somewhat misleading title and abstract and some use of language in the Discussion that seem to be claiming results that are not supported.

Response to reviewer #4:

Thank you for your comments. Please see below our point by point response to your comments:

- We have defined primary endpoint in the protocol as follows:

The primary endpoint is safety of tofacitinib in SLE subjects. In order to assess safety, toxicity is defined as any study drug-related Grade 3 adverse event or higher (as measured by the National Cancer Institute (NCI), Common Terminology Criteria for Adverse Events (CTCAE), Version 4.0).

- We agree with the reviewers that our definition of flare in this early phase study is truncated version of what have been traditionally used in large scale phase III trials. However, we still consider that any worsening of disease activity would have been captured by our definition. The subjects were followed up every week (face to face visits alternating with telephone visits) so the chance of missing a SLE flare is extremely low. The SLE disease flare was defined in our protocol as follows:

SLE disease flares will be defined as:

- Mild to moderate flare:
 - an increase in SLEDAI 2K score ≥ 3 but the total score is < 12 or
 - an increase in the PGA > 1 but the total score is < 2.5
- Severe Flare:
 - a SLEDAI 2K score > 12
 - a PGA > 2.5

- We have revised the results section to show data about primary and secondary outcomes:

There were 71 adverse events: 43 in the tofacitinib group and 28 in the placebo group, with no serious AEs in the tofacitinib group (Table 2). Most of the AEs observed in the tofacitinib group were mild (16/43) and moderate (5/43) upper respiratory infections that either self-resolved or after treatment with oral antibiotics. No herpes zoster reactivation, BK viremia, or venous thromboembolic events were recorded. There were no clinical or statistically significant changes observed in the tofacitinib group compared to baseline measurements and to the placebo group in other laboratory safety parameters (Table 3). As compared to placebo in the tofacitinib group, the hemoglobin difference in change score was -0.33 (95% CI -0.33,-0.88) at day 56, -0.20 (95% CI, -0.81,0.40) at day 84; white blood cell count difference in change score was -0.63(95% CI -0.63,-1.46) at day 56, -0.52 (95% CI, -1.35,0.30) at day 84; absolute neutrophil count difference in change score was -0.58(95% CI -0.58,-1.35) at day 56, -0.21 (95% CI, -0.85,0.43) at day 84; platelet count difference in change score was -15.36(95% CI -15.36,-35.79) at day 56, 5.3 (95% CI, -12.33,22.93) at day 84; serum AST difference in change score was 0.71(95% CI, 0.71,-5.14) at day 56, -13.8 (95% CI, -34.91,7.31) at day 84; and serum ALT difference in change score was 1.58(95% CI, 1.58,-4.55) at day 56, -2.15 (95% CI, -8.86,4.56) at day 84 (Table 3). None of these differences were statistically significant. None of the patients in either group met the disease flare criteria during the trial and there were no new BILAG2004 A or B scores. The baseline mean SLEDAI 2K score in the tofacitinib group was 5.1 ± 2.2 mean \pm standard deviation, compared to 5.5 ± 3.7 in the placebo group; the difference in change scores (tofacitinib vs. placebo) was 0.04 (95 % CI: -1.04,1.11) at day 56 and -0.72(95% CI: -1.98,0.53) at day 84. The baseline mean BILAG 2004 score in the tofacitinib group was 7.6 ± 4.6 mean \pm standard deviation, compared to 9.3 ± 4.3 in the placebo group; the difference in change scores (tofacitinib vs. placebo) was 1.56 (95 % CI: -1.86, 4.98) at day 56 and -2.04 (95% CI: -4.96,0.89) at day 84. The baseline mean PGA score in the tofacitinib group was 2.5 ± 2.8 mean \pm standard deviation, compared to 3.9 ± 2.8 in placebo group; the difference in change scores (tofacitinib vs. placebo) was 0.60 (95 % CI: -0.93, 2.13) at day 56 and 0.76 (95% CI: -1.01, 2.53) at day 84. None of these differences were statistically significant. The SLE serological disease activity (complement C3

and C4 levels) and the patient reported outcomes (SF-36) were similar at baseline visit and did not have a significant difference in change scores at day 56 or day 84 (Table 4). This study was not powered to assess clinical efficacy.

- We agree with the reviewer about the exploratory premise and findings of our study. We have mentioned in discussion and in conclusion about the limitations of a short duration trial and toned down the language accordingly. Our hope is that our report will serve as a stimulus for scientific community to explore further the cardiometabolic effects of tofacitinib in SLE.
- We have revised the abstract to more clearly mention primary, secondary outcomes and the exploratory analysis as below:

The primary outcome was to determine safety and tolerability of tofacitinib in subjects with mild-moderate SLE disease activity. The secondary outcomes included assessment of clinical response, as measured by the Systemic Lupus Erythematosus Disease Activity Index 2000 (SLEDAI-2K), physician global index (PGA), the British Isles Lupus Assessment Group disease activity index (BILAG 2004), and effects on quality of life measures (Short Form Health Survey (SF 36) and Multidimensional Assessment of Fatigue questionnaire (MD-Fatigue). In addition, we performed assessments of large, medium and small vascular function, measured plasma lipoproteins, neutrophil extracellular traps (NETs), immune cell subsets and their gene expression, to explore if Jakinibs improve cardiometabolic and immunologic parameters associated with enhanced CVD risk in SLE.

- We agree with the reviewer about the primary and secondary outcomes table being in supplementary data and moved them to the main manuscript, and have added the following statement to the methods:

The primary outcome of the study was defined as comparing rates of adverse events and rates of SLE disease flares between the tofacitinib group and the placebo group. The SLE disease flare was defined as an increase in SLEDAI 2K score of ≥ 3 or an increase in PGA >1 . The secondary outcomes included assessment of clinical response, effects on quality of life measures and several exploratory mechanistic studies to evaluate effect of the drug on immune dysregulation and cardiometabolic signatures associated with the development of premature cardiovascular disease.

- We have modified the abstract and manuscript based on the comments from the reviewer. The sentence mentioning meeting primary outcome in the abstract is replaced by the following statement, and the mention of meeting the secondary endpoints is deleted from the abstract.

No safety signals were detected during the study.

- We made following changes in the abstract and the manuscript to incorporate comments about conclusions:

In abstract:

Some of these improvements were more robust in SLE subjects with *STAT4*-risk allele.

In the Discussion section of the manuscript:

Results from the current trial suggest the effect of JAK inhibition was more robust in subjects with STAT4 risk allele, which is associated with a more severe SLE, and an increased risk of CV events.

- We added the following statements in the result section based on the comments from the reviewer:

There were significantly more African American subjects in the *STAT4* risk allele negative subgroup in both tofacitinib and placebo groups ($p=0.002$). The subjects with *STAT4* risk allele were younger and mostly Hispanic, but these differences were not statistically significant.

- We added the following statements in the discussion section based on the comments from the reviewer:

The JAK-STAT pathway is involved in intracellular signaling of multiple cytokines; therefore, additional mechanistic studies are needed to better characterize the pathways responsible for findings in this study.

- The fold changes in pSTAT1 in stimulated/unstimulated CD4⁺ cells in subjects on tofacitinib was significantly lower at day 14 and day 56 with return to baseline at day 84. These changes reflect the expected, known biological effect of tofacitinib based on studies from our group and others. There is indeed a downward but not significant trend in placebo group with no plausible explanation. We plan to further explore the pharmacodynamic effects of tofacitinib in much larger populations. We have added the following to results:

There was also a trend observed towards nonsignificant inhibition of pSTAT1 in subjects on placebo that is of unknown significance and without any known plausible biological explanation.

- We have added the following sentence in discussion:

There was a non-significant trend towards lower absolute neutrophil counts in subjects on tofacitinib.

- We have removed the reference to meeting primary endpoint as noted earlier. In addition, the following statement was added in the discussion:

As expected, there were more mild and moderate infections (mostly upper respiratory tract infections) in the tofacitinib group as compared to placebo.

- We have added the BILAG 2004 data to supplementary table 3 (now moved to main manuscript-table 4) and revised the safety and disease activity part of the results section based on the above comments from the reviewer, in addition have added the reference cited by the reviewer:

There were 71 adverse events: 43 in the tofacitinib group and 28 in the placebo group, with no serious AEs in the tofacitinib group (Table 2). Most of the AEs observed in the tofacitinib group were mild (16/43) and moderate (5/43) upper respiratory infections that either self-resolved or after treatment with oral antibiotics. No herpes zoster reactivation, BK viremia, or venous thromboembolic events were recorded. There were no clinical or statistically significant changes observed in the tofacitinib group compared to baseline measurements and to the placebo group in other laboratory safety parameters (Table 3). As compared to placebo in the tofacitinib group, the

hemoglobin difference in change score was -0.33 (95% CI -0.33,-0.88) at day 56, -0.20 (95% CI, -0.81,0.40) at day 84; white blood cell count difference in change score was -0.63(95% CI -0.63,-1.46) at day 56, -0.52 (95% CI, -1.35,0.30) at day 84; absolute neutrophil count difference in change score was -0.58(95% CI -0.58,-1.35) at day 56, -0.21 (95% CI, -0.85,0.43) at day 84; platelet count difference in change score was -15.36(95% CI -15.36,-35.79) at day 56, 5.3 (95% CI, -12.33,22.93) at day 84; serum AST difference in change score was 0.71(95% CI, 0.71,-5.14) at day 56, -13.8 (95% CI, -34.91,7.31) at day 84; and serum ALT difference in change score was 1.58(95% CI, 1.58,-4.55) at day 56, -2.15 (95% CI, -8.86,4.56) at day 84 (Table 3). None of these differences were statistically significant. None of the patients in either group met the disease flare criteria during the trial and there were no new BILAG2004 A or B scores. The baseline mean SLEDAI 2K score in the tofacitinib group was 5.1 ± 2.2 mean \pm standard deviation, compared to 5.5 ± 3.7 in the placebo group; the difference in change scores (tofacitinib vs. placebo) was 0.04 (95 % CI: -1.04,1.11) at day 56 and -0.72 (95% CI: -1.98,0.53) at day 84. The baseline mean BILAG 2004 score in the tofacitinib group was 7.6 ± 4.6 mean \pm standard deviation, compared to 9.3 ± 4.3 in the placebo group; the difference in change scores (tofacitinib vs. placebo) was 1.56 (95 % CI: -1.86, 4.98) at day 56 and -2.04 (95% CI: -4.96,0.89) at day 84. The baseline mean PGA score in the tofacitinib group was 2.5 ± 2.8 mean \pm standard deviation, compared to 3.9 ± 2.8 in placebo group; the difference in change scores (tofacitinib vs. placebo) was 0.60 (95 % CI: -0.93, 2.13) at day 56 and 0.76 (95% CI: -1.01, 2.53) at day 84. None of these differences were statistically significant. The SLE serological disease activity (complement C3 and C4 levels) and the patient reported outcomes (SF-36) were similar at baseline visit and did not have a significant difference in change scores at day 56 or day 84 (Table 4). This study was not powered to assess clinical efficacy.

- We added the following statement under results section in response to comments from the reviewer:

Despite randomization, there were some differences between tofacitinib and placebo group at the baseline visit. The subjects on placebo reported a statistically significant higher degree of fatigue, as measured by MD-Fatigue scale at baseline and a significant improvement in fatigue during the study. The subjects on placebo had higher baseline DAS-28-ESR and lower baseline anti-ds-DNA levels as compared to tofacitinib group, but these were not statistically significant differences (Table 4).

- The PGA was on a Likert scale of 0-3. We appreciate the reviewer for pointing this out and have corrected the data analysis and made changes to the Table 4 accordingly.
- We added the following statement in method to define DAS 28 used:

The SLE disease activity was determined using SLEDAI 2K, Disease Activity Score 28-Erythrocyte Sedimentation Rate (DAS-28-ESR), Physician Global Assessment (PGA) (Likert scale 0-3), and patient-reported outcomes (SF-36, Multidimensional Assessment of Fatigue questionnaire).

- We agree with the reviewer's comments and are planning to study the specific role of tofacitinib in modulation cardiovascular risk in SLE in a larger population over a longer time period. We believe keeping the clinical and cardiovascular findings together in one paper provides a much comprehensive picture to our intended reader. We have made several

changes in the abstract and throughout the manuscript to underscore the preliminary and exploratory aspect of our findings.

- We made several changes in the abstract and discussion to provide a more clear context of the study findings.

Reviewers' Comments:

Reviewer #1:

Remarks to the Author:

This paper continues to be unconvincing.

1. The first and second point raised in the previous review are still not sufficiently answered. While the authors' response that this was a safety study is fair enough, there is not the slightest indication of efficacy which should have been seen after 3 months. What does one care about a drug that may be safe in SLE if it is not efficacious? While the patients had stable disease, there was still some activity and that did not change at all compared with placebo. Also, baricitinib was already studied in a much larger number of patients with somewhat higher activity, so why was tofacitinib not assessed in such patients?
2. More importantly, the authors point toward modulation of immune and cardiometabolic parameters. However, none of the immune parameters assessed has ever been proven to be responsible for the pathogenesis of SLE and the ones that do play a role, namely anti-dsDNA autoantibodies, have not been assessed or data not shown.
3. With respect to the cardiometabolic aspects which are addressed at length and are also included in the title, the authors respond to the critique of lack of novelty by mentioning HDL and LCAT (as shown in Fig. 3). However, very similar data on tofacitinib effects have already been published many years ago in other diseases, such as by Wolk et al in *J Clin Lipidol* 2017; 11:1243 for psoriasis and by Charles-Schoeman et al in *Semin Arthritis Rheum* 2016; 46:71 for rheumatoid arthritis. Are the authors implying that SLE patients would behave differently than patients with other chronic diseases when receiving a jakinib? And why are the authors suggesting these data are novel if numerous papers have assessed the effects of Jakinibs on lipids in various diseases, none of which is cited (or did this reviewer miss it?).
4. Finally, the fact that the changes in pSTAT1 on placebo have no plausible explanation suggest that this may be a chance occurrence. But this then also raises the question regarding the changes on tofacitinib, irrespective of the fact that one would be surprised if a JAKinib would not have an effect on pSTAT1, as has also been often shown in other diseases.
5. In the first paragraph of the discussion the authors suggest that "jakinibs could modulate SLE vasculopathy and potentially mitigate CV risk". This is highly speculative and raises expectations that this study does not provide for the following reasons: it is a small study; it is not an outcomes study; disease activity did not change; and the whole issue of thromboembolic events with tofacitinib has not even been touched upon.

Reviewer #2:

Remarks to the Author:

I have been asked to follow up on the responses to reviewer 2 of the original manuscript.

The authors have done much to allay the concerns around the reporting of statistics and p-values and there is much more clarity regarding this. This is welcomed.

I would make the following further points (some reinforcing the original comments):

1. Given the primary outcome was safety and tolerability of Tofacitinib, and there was no power analysis in support of any outcomes, the title remains misleading. Safety and tolerability should be given more prominence as this was the purpose of the study. This is picked up in the discussion, but to claim "Tofacitinib modulates..." based on uncontrolled exploratory analysis is overstating the results which would need to be verified.
2. Whilst the revisions make clearer that the primary outcome is safety and tolerability, this is still downplayed in the paper with much more discussion of the secondary exploratory outcomes. It is interesting also that there is no attempt to statistically compare AE rates between arms, given the

array of results displayed for other outcomes. Were any pre-specified criteria for safety declared? Likewise tolerability appears to be undefined.

3. The authors are upfront that no formal power analysis was undertaken. However, it would be possible to give the level of precision that 20 vs 10 participants would give to the estimated rates, and therefore to their comparison.

4. I would not (personally) advocate the use of p-values to compare baseline measures. As the sample is randomised, any differences will necessarily arise due to chance which rather undermines them! In any case it is not clear whether the results are compared between the active and control groups overall, or between the stratified groups.

5. In light of the above comments, the discussion should be clearer about what can definitively be inferred from this study and what further explorations may be necessary to confirm the results.

Like reviewer 2, I also support that the study has been undertaken according both to the protocol and SAP. My comments are purely with regard to the presentation of results and ensuring the interpretation is consistent with the study aims and design.

Reviewer #3:

Remarks to the Author:

The authors have amended the manuscript according to the reviewers' comments.

Reviewer #4:

Remarks to the Author:

The authors have largely addressed the concerns of reviewer 4, with one remaining item (unless I am misreading). While the definition of the PGA has been corrected (Likert scale of 0-3), the results reported in the manuscript appear to remain reported on the original scale (mean PGA 3.9 in the placebo group).

Response to Reviewers:

On behalf of all authors of the manuscript thank you for giving us the opportunity to resubmit the revised manuscript. Please see below our point by point response and the manuscript have been with all changes highlighted in yellow.

REVIEWER COMMENTS

Reviewer #1 (Remarks to the Author):

This paper continues to be unconvincing.

1. The first and second point raised in the previous review are still not sufficiently answered. While the authors' response that this was a safety study is fair enough, there is not the slightest indication of efficacy which should have been seen after 3 months. What does one care about a drug that may be safe in SLE if it is not efficacious? While the patients had stable disease, there was still some activity and that did not change at all compared with placebo. Also, baricitinib was already studied in a much larger number of patients with somewhat higher activity, so why was tofacitinib not assessed in such patients?
2. More importantly, the authors point toward modulation of immune and cardiometabolic parameters. However, none of the immune parameters assessed has ever been proven to be responsible for the pathogenesis of SLE and the ones that do play a role, namely anti-dsDNA autoantibodies, have not been assessed or data not shown.
3. With respect to the cardiometabolic aspects which are addressed at length and are also included in the title, the authors respond to the critique of lack of novelty by mentioning HDL and LCAT (as shown in Fig. 3). However, very similar data on tofacitinib effects have already been published many years ago in other diseases, such as by Wolk et al in J Clin Lipidol 2017; 11:1243 for psoriasis and by Charles-Schoeman et al in Semin Arthritis Rheum 2016; 46:71 for rheumatoid arthritis. Are the authors implying that SLE patients would behave differently than patients with other chronic diseases when receiving a jakinib? And why are the authors suggesting these data are novel if numerous papers have assessed the effects of Jakinibs on lipids in various diseases, none of which is cited (or did this reviewer miss it?).
4. Finally, the fact that the changes in pSTAT1 on placebo have no plausible explanation suggest that this may be a chance occurrence. But this then also raises the question regarding the changes on tofacitinib, irrespective of the fact that one would be surprised if a JAKinib would not have an effect on pSTAT1, as has also been often shown in other diseases.
5. In the first paragraph of the discussion the authors suggest that "jakinibs could modulate SLE vasculopathy and potentially mitigate CV risk". This is highly speculative and raises expectations that this study does not provide for the following reasons: it is a small study; it is not an outcomes study; disease activity did not change; and the whole issue of thromboembolic events with tofacitinib has not even been touched upon.

Response to reviewer #1:

Thank you for your comments. Please see below our point by point response to your comments:

1. The study was designed as an early phase study to assess the safety of tofacitinib and was started before the baricitinib study, a multicenter study with much larger number of patients recruited (N=314), longer duration (24 weeks), and more active subjects (mean SLEDAI 2K scores between 8-9 and 40% patients had SLEDAI 2K >10) was published. In our study the baseline mean SLEDAI 2K scores in the tofacitinib group was 5.1 ± 2.2 (mean \pm standard deviation). Whether tofacitinib is effective in a larger group of SLE patients with higher disease

activity needs to be determined by future studies. Again, we want to emphasize that the purpose of our study was not to assess efficacy and, as such, the study was designed to address safety, but also to perform a number of mechanistic studies focusing on cardiometabolic parameters that were not addressed in the baricitinib study. Therefore, the studies in our view complement each other.

2. We respectfully disagree with the reviewer. Many studies indicate that the type I IFN pathway and neutrophil dysregulation play putative pathogenic roles in SLE, as confirmed by the anifrolumab trials in humans and many murine studies. Specifically, our group and others have found that a dysregulated neutrophil/type I IFN axis is pathogenic in vascular disease in SLE and have published extensively on this subject. Our study showed improvement in 3 parameters of the innate immune system; downregulation of Interferon Stimulated Genes, reduction in numbers of low-density granulocytes, and reduction of circulating NET complexes. The data about anti-ds-DNA antibodies are shown in Table 1 row 18 and Table 3 row 10. There was a numerical increase in anti-ds-DNA antibody titers in both tofacitinib and placebo groups which was not statistically significant. Furthermore, whether anti-dsDNA antibodies do indeed play a pathogenic role in SLE or are mere diagnostic markers that associate with disease activity and renal disease, remains to be systematically determined. The study did not assess other autoantibodies, but we hope future studies in sicker patients will assess prospectively and systematically the modulation of T and B cell responses in SLE. In this patient population with mild/moderate disease, gene expression and flow cytometry analysis pointed toward modulation of innate immune responses. Again, a strength of this paper is the mechanistic studies, which were not included in the baricitinib paper.

We made the following revisions in the manuscript, following the comments of the reviewer, for further clarification:

In the introduction section:

The role of the innate immune system, specifically type I interferons, low density neutrophils and neutrophil extracellular traps (NETs) are now recognized as potential fundamental players in SLE pathogenesis and its associated vascular damage(2).

In the results section:

The SLE serological disease activity (anti-ds-DNA, complement C3 and C4 levels) and the patient reported outcomes (SF-36) were similar at baseline visit and did not have a significant difference in change scores at day 56 or day 84 (Table 4). There was a numerical increase in anti-ds-DNA antibody titers in both groups during the study, but these increases were not statistically significant.

And

The subjects on placebo had higher baseline DAS-28-ESR and lower baseline anti-ds-DNA levels as compared to tofacitinib group, but these were not statistically significant differences (Table 4).

In the discussion section:

Previous work from our group and others has implicated a pathophysiologic alliance between type I IFNs, LDGs, and enhanced NET formation as a mechanism that promotes premature atherosclerosis and vasculopathy in SLE

3. The novel findings from the current study are improvement in all three major determinants of cardiovascular risk related to lipid metabolism, i.e, cHDL levels, HDL particle size, LCAT concentration and cholesterol efflux capacity, without an increase in triglycerides or LDL in patients with SLE. The previous studies (such as the ones cited by the reviewer and others cited in our manuscript) have reported some of these effects of tofacitinib but with worsening of other parameters of lipid metabolism. The favorable effects of tofacitinib on lipid metabolism without a concomitant worsening is shown for the first time in our study. Furthermore, no such data is reported in lupus patients whose response to tofacitinib may be different than patients with other autoimmune diseases and where dysregulation in HDL appears to play an important pathogenic role. Not only did we find an improvement in HDL levels but also in its function and to our knowledge this is also novel. We have added both citations mentioned by the reviewer in the references.
4. We have reported the data about changes in pSTAT1 and other effects of tofacitinib in SLE patients for the first time. These data need to be confirmed by another study in a larger population. The placebo responses of biologic parameters are not uncommon in SLE but there was clear target engagement by tofacitinib based on pSTAT data.
5. In our study, no thromboembolic events were observed. The study was short duration and not designed to address this potential issue. Furthermore, the concerns with thromboembolism are reported primarily with the higher dose (10 mg) of tofacitinib. The language in the discussion section as mentioned by the reviewer is written as thought-provoking statements and to stimulate intellectual discourse. We believe that our intent is clearly reflected in the following statement at the end of the first paragraph in the discussion section:

The observation that jakinibs could modulate SLE vasculopathy and potentially mitigate CV risk could have important implications in this patient population.

Reviewer #2 (Remarks to the Author):

I have been asked to follow up on the responses to reviewer 2 of the original manuscript.

The authors have done much to allay the concerns around the reporting of statistics and p-values and there is much more clarity regarding this. This is welcomed.

I would make the following further points (some reinforcing the original comments):

1. Given the primary outcome was safety and tolerability of Tofacitinib, and there was no power analysis in support of any outcomes, the title remains misleading. Safety and tolerability should be given more prominence as this was the purpose of the study. This is picked up in the discussion, but to claim "Tofacitinib modulates..." based on uncontrolled exploratory analysis is overstating the results which would need to be verified.
2. Whilst the revisions make clearer that the primary outcome is safety and tolerability, this is still downplayed in the paper with much more discussion of the secondary exploratory outcomes. It is interesting also that there is no attempt to statistically compare AE rates between arms, given the array of results displayed for other outcomes. Were any pre-specified criteria for safety declared? Likewise tolerability appears to be undefined.

3. The authors are upfront that no formal power analysis was undertaken. However, it would be possible to give the level of precision that 20 vs 10 participants would give to the estimated rates, and therefore to their comparison.
4. I would not (personally) advocate the use of p-values to compare baseline measures. As the sample is randomised, any differences will necessarily arise due to chance which rather undermines them! In any case it is not clear whether the results are compared between the active and control groups overall, or between the stratified groups.
5. In light of the above comments, the discussion should be clearer about what can definitively be inferred from this study and what further explorations may be necessary to confirm the results.

Response to reviewer #2:

Thank you for your comments. Please see below our point by point response to your comments:

1. Our study was designed to assess safety and to do exploratory mechanistic study from the outset as can be inferred from the title of our protocol "*Safety of tofacitinib, an oral Janus kinase inhibitor, in Systemic Lupus Erythematosus; a Phase Ib clinical trial and associated mechanistic studies*". We purposefully designed the study to further explore the cardiometabolic effects of inhibiting JAK STAT pathway in lupus based on our work done on the animal model of lupus (Furumoto Y, Smith CK, Blanco L, Zhao W, Brooks SR, Thacker SG, et al. Tofacitinib Ameliorates Murine Lupus and Its Associated Vascular Dysfunction. *Arthritis Rheumatol.* 2017;69(1):148-60).

However, since tofacitinib was never systematically studied in human lupus before our study we wanted to keep the safety and tolerability as the primary outcome. The methods and results sections clearly state the primary outcome of the study as safety and tolerability of tofacitinib in subjects with mild-moderate SLE. The title of the manuscript reflects the findings that would stimulate intellectual curiosity and invite further exploration of the idea that immune modulation can effect cardiometabolic outcomes. In addition to what is stated in the methods and results section we have revised the start of discussion section as follows to more prominently state safety and tolerability as the primary outcome of the study.

Discussion:

In this study, short-term use of tofacitinib in subjects with mild-to-moderate SLE was overall safe, well tolerated, with no unexpected AEs, thromboembolic events or opportunistic infections. Our exploratory analyses showed that tofacitinib led to significant positive modulation of cardiometabolic and immunologic parameters previously linked to increased coronary atherosclerotic plaque, vascular inflammation, and abnormalities in blood vessel function in lupus and in the general population(24, 25).

2. Most of the AE were comparable in both groups and there was an expected increase in mild and moderate infection related AE in the tofacitinib group. We have added the comparison of the two groups to the adverse events table (Table 2). The descriptive statistics about details of AEs are also mentioned in the manuscript. The pre-specified safety criteria in the protocol were: **In order to assess safety, toxicity is defined as any study drug-related Grade 3 adverse event or higher (as measured by the National Cancer Institute (NCI), Common Terminology Criteria for Adverse Events (CTCAE), Version 4.0). Grade 3 adverse**

events will include measures of standard laboratory tests including serum chemistries, urinalysis, complete blood counts and lipid profiles at screening, baseline and conclusion of the treatment period and at the end of the study. There were no concerns about tolerability, none of the subject discontinued intervention as shown in the CONSORT flow diagram.

We have added the following statement in the manuscript under results section:

There were 71 adverse events: 43 in the tofacitinib group and 28 in the placebo group, with no serious AEs in the tofacitinib group; the differences in AE were not statistically significant (Table 2).

3. Thank you for the suggestion. Given high level, of heterogeneity in this population, we would like to refrain from reporting level of precision. However, we did report the confidence intervals for the comparisons of the outcome measures. Such information and the data from this study can be used to assist power calculations in future studies.
4. We agree that in a randomized study the baseline characteristics are due to chance. We have added p-values to baseline characteristics to respond to comments received previously by a reviewer. The results were compared between the active and control groups overall and between the stratified groups, but we only reported results for the stratified groups.
5. We made revisions in the discussion section as noted above in our response to comment # 1.

Reviewer #3 (Remarks to the Author):

The authors have ammended the manuscript according to the reviewers' comments.

Response

Thank you very much.

Reviewer #4 (Remarks to the Author):

The authors have largely addressed the concerns of reviewer 4, with one remaining item (unless I am misreading). While the definition of the PGA has been corrected (Likert scale of 0-3), the results reported in the manuscript appear to remain reported on the original scale (mean PGA 3.9 in the placebo group).

Response

Thank you for pointing out the discrepancy in PGA results. The reported PGA results were left over from the previous version and it is now revised and reported correctly. We made the following corrections:

In the manuscript under results:

The baseline mean PGA score in the tofacitinib group was 0.8 ± 0.8 mean (± standard deviation), compared to 1.2 ± 0.9 in placebo group; the difference in change in scores (tofacitinib vs. placebo) was 0.18 (95 % CI: -0.28, 0.64) at day 56 and 0.23 (95% CI: -0.30, 0.76) at day 84.

Table 4

PGA	0.8 ±	0.8 ±	0.9 ±	1.2 ±	0.9 ±	0.8 ±	0.21	0.43	0.38
	0.8	0.9	0.7	0.9	0.8	0.8		0.18 (-0.28,	0.23 (-0.30,

Reviewers' Comments:

Reviewer #1:

Remarks to the Author:

The authors of this manuscript continue to provide insufficiently substantiated claims and the revised version of the paper did not address the critique raised previously clearly. Here are some additional comments.

A. Autoantibodies:

With the provision of new data, we learn that only ~20% (!) had anti-dsDNA antibodies while ~40% had a lupus anticoagulant. Usually SLE patients have ~50% anti-DNA and ~10% LA. Thus, this population is not very representative of common SLE patient populations – a new point that is worrisome.

Also, the data on anti-dsDNA antibody levels (why were they not shown in the original paper?) further support this reviewer's notion on the inefficacy of tofacitinib in this study, since anti-DNA increased by about 60% during treatment – about the same extent as seen in the placebo group. Of course, given the very small number of anti-DNA positive patients, this increase was not significant, but the trend was clearly there.

It is also surprising to hear that "the study did not assess other autoantibodies" – every single lupus center in the world has data on anti-Sm, RNP, Ro, and La for all their patients (and once present they are always present, contrasting anti-dsDNA); were all of these negative or very infrequent and therefore not shown?

B. Pathogenetic role of type I IFN pathway:

While the authors "respectfully disagree" with the reviewer's notion on the lacking evidence of a pathogenetic role of IFN in SLE, they themselves speak of "putative pathogenetic roles" in SLE ! So is it pathogenic or just "putatively pathogenic"?

Here some additional food for thought: The mere fact that baricitinib changed the interferon signature but with no relationship to clinical improvement simply suggests that IFN upregulation is not involved in pathogenesis of SLE but a mere epiphenomenon. Further support against a major role comes, indeed, from the anifrolumab study mentioned by the authors in their response: among patients with a high IFN gene signature the response rate was 48% while it was 47% in those with a low signature – 1% difference. And a second anifrolumab trial had a negative result. Don't the authors know all these data? Why do they insist on providing this putatively false information? It is well established that nucleic acids containing autoantigens and their immune complexes can activate nucleic acid sensing TLRs and then in turn drive type I IFN production (see Ann Marshak-Rothstein's work). And immune complexes can do much more harm than just activating IFN pathways.

Overall, the authors' observation that tofacitinib changed the IFN signature is in line with the known baricitinib data, which had no association at all with clinical efficacy (see above).

C. Coming back to the data of the paper:

LDGs: The authors use Fig. 2a to claim that tofacitinib decreased LDGs significantly compared with placebo. However, looking at Fig. 2a, one can only see a significant increase of LDGs in the placebo population but anything but a significant decrease of LDGs from baseline in the tofacitinib group (overlapping confidence intervals). How can they claim a reduction of LDGs by tofacitinib? Rather than making this claim, they should ask why there was an increase in LDGs in the placebo population – is the method sufficiently reproducible? Or does it simply have to do with the similar

changes seen for neutrophil counts (increase between D1 and 56 with PL and small decrease with tofa over time), for which the authors state that there was no change? Very, very confusing and simply not in line with what one sees! Statistics are just not the holy grail when such changes occur in a placebo group without any known reason.

NETs and STAT4: while there is a statistically significant difference between the groups in Fig. 4c, the overlap between the groups is huge. Only 3 STAT4+ve pts have higher NET levels than the STAT4-ve ones (Fig. 4c). Again, the placebo group, whether STAT4+ or negative, has a much lower NET expression at baseline (Figure 4d)! How can one draw any statistical conclusions with such baseline differences and so few patients, with an imbalance in numbers between PL and Active treatment arms?

The other points raised before (e.g. regarding lipid data) have also not been sufficiently answered.

Reviewer #2:

Remarks to the Author:

The authors have adequately addressed the remaining concerns I raised.

Reviewer #4:

Remarks to the Author:

My concern has been addressed and I have no additional comments.

Response to Reviewer:

On behalf of all authors of the manuscript thank you for giving us the opportunity to resubmit the revised manuscript. Please see below our point by point response and the manuscript have been with all changes highlighted in yellow.

REVIEWER COMMENTS

Reviewer #1 (Remarks to the Author):

The authors of this manuscript continue to provide insufficiently substantiated claims and the revised version of the paper did not address the critique raised previously clearly. Here are some additional comments.

A. Autoantibodies:

With the provision of new data, we learn that only ~20% (!) had anti-dsDNA antibodies while ~40% had a lupus anticoagulant. Usually SLE patients have ~50% anti-DNA and ~10% LA. Thus, this population is not very representative of common SLE patient populations – a new point that is worrisome.

Also, the data on anti-dsDNA antibody levels (why were they not shown in the original paper?) further support this reviewer's notion on the inefficacy of tofacitinib in this study, since anti-DNA increased by about 60% during treatment – about the same extent as seen in the placebo group. Of course, given the very small number of anti-DNA positive patients, this increase was not significant, but the trend was clearly there.

It is also surprising to hear that “the study did not assess other autoantibodies“ – every single lupus center in the world has data on anti-Sm, RNP, Ro, and La for all their patients (and once present they are always present, contrasting anti-dsDNA); were all of these negative or very infrequent and therefore not shown?

B. Pathogenetic role of type I IFN pathway:

While the authors “respectfully disagree“ with the reviewer's notion on the lacking evidence of a pathogenetic role of IFN in SLE, they themselves speak of “putative pathogenetic roles“ in SLE ! So is it pathogenic or just “putatively pathogenic“?

Here some additional food for thought: The mere fact that baricitinib changed the interferon signature but with no relationship to clinical improvement simply suggests that IFN upregulation is not involved in pathogenesis of SLE but a mere epiphenomenon. Further support against a major role comes, indeed, from the anifrolumab study mentioned by the authors in their response: among patients with a high IFN gene signature the response rate was 48% while it was 47% in those with a low signature – 1% difference. And a second anifrolumab trial had a negative result. Don't the authors know all these data? Why do they insist on providing this putatively false information? It is well established that nucleic acids containing autoantigens and their immune complexes can activate nucleic acid sensing TLRs and then in turn drive type I IFN production (see Ann Marshak-Rothstein's work). And immune complexes can do much more

harm than just activating IFN pathways.

Overall, the authors' observation that tofacitinib changed the IFN signature is in line with the known baricitinib data, which had no association at all with clinical efficacy (see above).

C. Coming back to the data of the paper:

LDGs: The authors use Fig. 2a to claim that tofacitinib decreased LDGs significantly compared with placebo. However, looking at Fig. 2a, one can only see a significant increase of LDGs in the placebo population but anything but a significant decrease of LDGs from baseline in the tofacitinib group (overlapping confidence intervals). How can they claim a reduction of LDGs by tofacitinib? Rather than making this claim, they should ask why there was an increase in LDGs in the placebo population – is the method sufficiently reproducible? Or does it simply have to do with the similar changes seen for neutrophil counts (increase between D1 and 56 with PL and small decrease with tofa over time), for which the authors state that there was no change? Very, very confusing and simply not in line with what one sees! Statistics are just not the wholly grail when such changes occur in a placebo group without any known reason.

NETs and STAT4: while there is a statistically significant difference between the groups in Fig. 4c, the overlap between the groups is huge. Only 3 STAT4+ve pts have higher NET levels than the STAT4-ve ones (Fig. 4c). Again, the placebo group, whether STAT4+ or negative, has a much lower NET expression at baseline (Figure 4d)! How can one draw any statistical conclusions with such baseline differences and so few patients, with an imbalance in numbers between PL and Active treatment arms?

The other points raised before (e.g. regarding lipid data) have also not been sufficiently answered.

Response:

- A. The data about the autoantibodies are cross-sectional at the time of enrollment. The anti-ds-DNA antibody levels fluctuate during the disease course and do not always correlate with clinical disease activity. Our long standing SLE Natural History Cohort (where the subjects for this clinical trial were recruited from) has ~60% anti-DNA positivity rate commensurable to what is described in other lupus cohorts. As mentioned in the manuscript, all subjects who participated in this trial fulfilled the 1997 revised ACR classification criteria for SLE. Furthermore, the prevalence of Lupus anticoagulant in SLE is reported to be between 11%-30% (Ünlü O, Zuilü S, Erkan D. *Eur J Rheumatol.* 2016;3(2):75-84. doi:10.5152/eurjrheum.2015.0085). Because we only enrolled patients with mild-moderate disease activity, this also may have limited the positivity of ds-DNA levels at the time of enrollment. We consider that our clinical trial population is representative of a typical lupus population; some of the differences as pointed out by the reviewers are due to small population and short duration of the trial. Lastly to reiterate, the trial was not designed to look at the efficacy of tofacitinib and cannot be compared to much larger trials of longer duration. We did not collect the data on anti-Sm, RNP, Ro, and La as part of this trial and its design. We do have data collected as part of the SLE Natural History protocol and the positivity rates are similar to what has been reported in the literature.

- B. The dysregulation of type I IFN in SLE pathogenesis is an evolving area of research, hence we mentioned it is putative. Nevertheless, there is overwhelming evidence from the literature that this pathway is dysregulated in SLE and studies that have depleted pDCs have shown evidence of efficacy in skin manifestations. Of course there are few who disagree and have shown some contradictory data but the overwhelming majority of lupus experts do agree on the role of aberrant type I IFN in SLE. Furthermore, several lupus susceptibility genes are related to the IFN pathways and up to 80% of SLE patients have increased expression of interferon stimulated genes in peripheral blood. In a recent article published in Nature Communications, Antonios Psarras et. al have extensively described the role of IFN in SLE (Psarras, A., Alase, A., Antanaviciute, A. et al. Functionally impaired plasmacytoid dendritic cells and non-haematopoietic sources of type I interferon characterize human autoimmunity. Nat Commun 11, 6149 (2020). <https://doi.org/10.1038/s41467-020-19918-z>). The clinical trials in lupus have shown mixed results in reference to IFN pathway which is indicative of the heterogeneity and challenges in treating SLE. The fact that IFN pathway was abrogated by tofacitinib in this trial indicates its target engagement but any role in clinical efficacy is yet to be determined.

To address the point made by the reviewer, we have added the following statement in the discussion section:

The exact role of type I IFN in the pathogenesis of SLE is still being defined and abrogating this pathway may not lead to decreases in disease manifestations, as evidence by mixed results from recent clinical trials using interferon receptor blocker Anifrolumab and the plasmacytoid dendritic cells specific receptor antibody BIIB059(33,34).

- C. Overall we would like to restate that this was an early phase study with the primary endpoint of safety of tofacitinib in SLE and assessment of modulation of putative cardiometabolic parameters. There were several exploratory mechanistic studies performed in this trial to understand the role of the JAK-STAT pathway and the effects resulting from blocking this pathway by tofacitinib. We presented the data for the scientific community to venture further on these findings. In reference to LDGs were reduced in the tofacitinib group. However, we do agree that there was an increase in % of circulating LDGs in placebo. On possible explanation is the expansion of LDG remained unchecked in the placebo while the treatment group responded by overall reduction in neutrophils as well as LDGs. The role of LDG and NETosis in the pathogenesis of SLE is increasingly being recognized by our group and many others. The other objections about NETs and STAT 4 as well as regarding lipid data are good discussion points raised by the reviewer. However, we have shown the data in the spirit of scientific advancement and an opportunity to bridge current knowledge gaps in SLE, nowhere in the manuscript we have claimed these to be the final verdict on SLE treatment.

To address these points we have made following changes in the manuscript:

In the results section under the heading, *Tofacitinib modulated dysregulated neutrophil responses*, the following statements were added:

There was also a concomitant increase in LDGs in placebo group which partially explain this statistically significant difference.

However, even though these differences were significant but there was an overlap between the groups and there was also a lower NET expression in placebo subjects at the baseline.

In the discussion section we have revised/added the following statements:

The observation that jakinibs may have a possible role in modulating SLE vasculopathy and potentially mitigating CV risk could have important implications in this patient population.

In addition, there were differences in the baseline between the groups and changes observed in placebo group which may be partially responsible for some the results observed in the secondary outcomes of this study.

Reviewers' Comments:

Reviewer #1:

Remarks to the Author:

While the data are still not convincing, the authors at least acknowledge limitations and incongruences in the revised version of the manuscript.

Response to Reviewers:

On behalf of all authors of the manuscript thank you for giving us the opportunity to resubmit the revised manuscript. Please see below our point by point response and the manuscript have been with all changes highlighted in yellow.

REVIEWER COMMENTS

Reviewer #1 (Remarks to the Author):

Reviewer #1 (Remarks to the Author):

While the data are still not convincing, the authors at least acknowledge limitations and incongruences in the revised version of the manuscript.

Response:

We would look to thank the reviewer for the feedback.

The revised manuscript is being submitted based on the editorial requests.